



# Reaching 1.5$^o$C and 2.0$^o$C global surface temperature targets using stratospheric aerosol geoengineering

Simone Tilmes[1], Douglas E. MacMartin[2], Jan T. M. Lenaerts[3], Leo van Kampenhout[4],
Laura Muntjewerf[5], Lili Xia[6], Cheryl S. Harrison[7], Kristen M. Krumhardt[8], Michael J. Mills[1],
Ben Kravitz[9,10], and Alan Robock[6]

[1]Atmospheric Chemistry, Observations, and Modeling Laboratory, National Center for Atmospheric Research, Boulder, CO, USA
[2]Mechanical and Aerospace Engineering, Cornell University, Ithaca, NY, USA
[3]Department of Atmospheric and Oceanic Sciences, University of Colorado Boulder, CO, USA
[4]Institute for Marine and Atmospheric Research, Utrecht University, Utrecht, The Netherlands
[5]Department of Geoscience and Remote Sensing, Delft University of Technology, The Netherlands
[6]Department of Environmental Sciences, Rutgers University, New Brunswick, NJ, USA
[7]Port Isabel Campus, University of Texas, Rio Grande Valley, USA
[8]Climate Global Dynamics Laboratory, National Center for Atmospheric Research, Boulder, CO, USA
[9]Department of Earth and Atmospheric Sciences, Indiana University, Bloomington, IN, USA
[10]Atmospheric Sciences and Global Change Division, Pacific Northwest National Laboratory, Richland, WA, USA

**Correspondence:** Simone Tilmes(tilmes@ucar.edu)

**Abstract.** We propose new testbed model experiments for the Geoengineering Model Intercomparison Project (GeoMIP) that are designed to limit global warming to 1.5$^o$C or 2.0$^o$C above 1850–1900 conditions using stratospheric aerosol geoengineering (SAG). The new modeling experiments use the overshoot scenario defined in CMIP6 (SSP5-34-OS) as a baseline scenario and are designed to reduce side effects of SAG in reaching three temperature targets: global mean surface temperature, and

inter-hemispheric and pole-to-equator surface temperature gradients. We further compare results to another SAG simulation using a high emission scenario (SSP5-85) as a baseline scenario in order to investigate the dependency of impacts using different injection amounts to offset different amounts of warming by SAG. The new testbed simulations are performed with the CESM2(WACCM6). We use a feedback algorithm that identifies the needed amount of sulfur dioxide injections in the stratosphere at four predefined latitudes, 30$^o$N, 15$^o$N, 15$^o$S, and 30$^o$S, to reach the three temperature targets. Here we ana-

lyze climate variables and quantities that matter for societal and ecosystem impacts. We find that changes from present day conditions (2015–2025) in some variables depend strongly on the defined temperature target (1.5$^o$C vs 2.0$^o$C). These include surface air temperature and related impacts, the Atlantic Meridional Overturning Circulation (AMOC), which impacts ocean net primary productivity, and changes in ice sheet surface mass balance, which impacts sea-level rise. Others, including global precipitation changes and the recovery of the Antarctic ozone hole, depend strongly on the amount of SAG application. Further-

more, land net primary productivity as well as ocean acidification depend mostly on the global atmospheric $CO_2$ concentration and therefore the baseline scenario. Multi-model comparisons of the experiments proposed here would help identify consequences of scenarios that include strong mitigation, carbon dioxide removal with some SAG application, on societal impacts and ecosystems.



## 1   Introduction

Large-scale mitigation efforts to phase out anthropogenic emissions are likely no longer sufficient to keep global mean surface temperature from rising less than $2^oC$ above pre-industrial levels, which is required to avoid significant impacts on societies and ecosystems (Masson-Delmotte et al., 2018). Stratospheric aerosol geoengineering (SAG) has been suggested as part of a portfolio of responses, including mitigation, adaptation, and carbon dioxide removal, to potentially reach required surface temperature targets and to reduce some of the effects of anthropogenic interference in the climate system (e.g., Long and
Shepherd, 2014; Lawrence et al., 2018; MacMartin et al., 2018). Here we present climate model experiments designed to assess impacts as a function of future greenhouse gas concentrations, the amount of SAG application, target temperatures, and the details of the application.

Various uniformly defined stratospheric aerosol geoengineering modeling experiments of different complexity have been designed within the Geoengineering Model Intercomparison Project (GeoMIP) to be performed by different modeling groups,
within Coupled Model Intercomparison Project 5 (CMIP5) (Kravitz et al., 2011) and CMIP6 (Kravitz et al., 2015). These simulations involve either injecting sulfur dioxide at the equator or using earlier derived prescribed aerosol distributions to reach the described goals (e.g., Pitari et al., 2014). These were designed, for instance, to keep the radiative forcing at 2020 levels, or apply a constant injection followed by a termination of the injection after 50 years. New GeoMIP experiments were designed for CMIP6, using a high forcing SSP5-85 scenario as a baseline and applying either sulfur dioxide injections or solar
dimming in order to reach the moderate radiative forcing of the SSP2-45 scenario (Kravitz et al., 2015). However, no Tier 1 GeoMIP experiments have been designed so far to achieve the $2.0^oC$ and $1.5^oC$ required temperature targets of the Paris Agreement. Furthermore, earlier GeoMIP experiments specify injections at or in a region around the equator, which result in excessive cooling of the tropics and less cooling of high latitudes, in turn causing large-scale precipitation shifts (Kravitz et al., 2013).

The geoengineering large ensemble (GLENS) project has defined experiments that aim to keep surface temperature values at close to present day levels to reduce impacts from global warming (Tilmes et al., 2018). The experiments used a feedback controller to maintain global average surface temperatures, as well as equator-to-pole and interhemispheric temperature gradients, at 2020 levels. GLENS was based on a high forcing future climate scenario (RCP8.5) and required an increasing amount of sulfur injection with time. GLENS simulations have shown that keeping global surface temperature and temperature gradients
changing results in benefits with respect to temperature related impacts compared to experiments that only focus on controlling for global surface temperature (Kravitz et al., 2019). However, there are other changes in the climate system that do not directly correlate with surface temperatures. Those include changes in atmospheric circulation and transport, monsoonal rainfall, and chemistry, as well as some responses of the biosphere on land and ocean. The magnitude of changes has been shown to be at least in part dependent on the applied amount and details of the application of SAG (Kravitz et al., 2017; Richter et al., 2018;
Kravitz et al., 2019; Simpson et al., 2019; MacMartin et al., 2019). Furthermore, risks to climate and ecosystems posed by a sudden SAG termination grow with increasing amount of sulfur injection. Consequently, side effects and risks depend strongly





on the required amount of intervention application, which is defined by the desired targets and the underlying greenhouse gas concentration pathway.

Several studies have pointed out that SAG may be able to reduce some of the effects of global warming temporarily while decarbonization efforts (including mitigation and negative emissions through carbon dioxide removal) are ramped up. A so-called peak-shaving scenario was proposed that would potentially help prevent reaching tipping points until greenhouse gas levels have been sufficiently reduced (Wigley, 2006; Tilmes et al., 2016; MacMartin et al., 2018; Lawrence et al., 2018). Tilmes et al. (2016), and Jones et al. (2018), have produced simulations that kept surface temperature increases to $1.5^oC$ or $2^oC$ levels using different RCP forcing scenarios. Jones et al. (2018) used the RCP2.6 scenario as a baseline, resulting in a slight reduction of temperature by the end of the 21st century. Their scenario therefore did not require continuously increasing injections around the equator, but lead to some injection reductions by the end of the century to reach $1.5^oC$ temperature targets. Tilmes et al. (2016) used a late decarbonization pathway, starting in 2040 from the high forcing scenario SSP8.5 and applied different amounts of stratospheric aerosol geoengineering to keep surface temperatures to $2.0^oC$ and $1.5^oC$, using a prescribed aerosol distribution scaled to produce the required cooling. Neither Jones et al. (2018) nor Tilmes et al. (2016) used a feedback algorithm or the multiple injection locations in their approach, as was done in GLENS, and their results showed continued warming in high latitudes and precipitation shifts, while reaching global temperature targets.

The new GeoMIP testbed experiment, proposed here, combines two main objectives that have only been addressed separately in previous studies. First, we apply a feedback controller to maintain three temperature targets, in order to reduce some of the side effects identified in earlier studies. Second, we use an overshoot scenario as the baseline scenario to limit the needed amount and duration of SAG to reach a $2.0^oC$ or $1.5^oC$ surface temperature target. To facilitate baseline scenarios that allow similar peak-shaving geoengineering experiments, as described by Tilmes et al. (2016), CMIP6 designed the overshoot scenario (OS) SSP5-34-OS (O'Neill et al., 2016). This scenario follows the high forcing scenario SSP5-85 until 2040 and then applies drastic decarbonization efforts, including mitigation and active carbon dioxide removal to produce net-negative emissions after 2070. The SSP5-34-OS scenario applies a sudden change in behavior in the consumption of fossil fuel emissions and also assumes large amounts of carbon removal. This produces a carbon dioxide ($CO_2$) concentration overshoot and a surface temperature profile that significantly overshoots the required temperature target before 2100.

We use the state-of-the-art Community Earth System Model version 2 (CESM2) with the Whole Atmosphere Community Climate Model (WACCM6) atmospheric component, from here on called WACCM6, which has been used for CMIP6 simulations. Section 2 describes the model as well as the experiments which are designed to reach $1.5^oC$ and $2.0^oC$ surface temperature targets. For baseline simulations we use the SSP5-85 and SSP5-34-OS scenarios to contrast differences that arise if applying SAG to reach the $1.5^oC$ temperature target. This comparison motivates the use of the overshoot scenario compared to the high forcing scenario. Resulting sulfur injections, stratospheric sulfur burden and comparisons of the efficiency using different scenarios is done in Section 3. The outcomes of these simulations are discussed in Section 4, where we summarize large-scale effects of SAG on surface temperature and precipitation, sea-surface temperatures and the Atlantic Meridional Overturning Circulation (AMOC). In addition, we include some diagnostics that are important for ecosystem and societal impact studies including changes in land primary productivity and land ice mass balance, effects on ocean ecosystems and the





recovery of the Antarctic ozone hole. We do not discuss any detailed regional outcomes based on one ensemble member and a single model. Some comparisons are performed to the GLENS project, to identify potential ranges of outcomes using an earlier CESM model version. Discussions and conclusions are presented in Section 5. The main goal of this paper is to estab-

lish a protocol for the new model experiments and motivate other modeling groups to conduct the same experiments, thereby allowing for an analysis of the outcomes from a multi-model perspective.

## 2 Model description and experimental design

### 2.1 Model description

The model experiments described here were performed with the WACCM6. Details on CESM2 and WACCM6 model configu-

rations, including an overview of the performance and new features, are described by Danabasoglu et al. (2019) and Gettelman et al. (2019), respectively. The WACCM6 atmospheric model uses a horizontal resolution of $1.25^{o}$ in longitude and $0.95^{o}$ in latitude, and 70 vertical layers, reaching up to 140 km height above sea level ($6 \times 10^{-6}$ hPa). Stratospheric dynamics perform well compared to observations, producing an interactive quasi-biennial oscillation (Gettelman et al., 2019). The simulations are performed with comprehensive tropospheric, stratospheric, mesospheric, and lower thermospheric (TSMLT) chemistry (Em-

mons, L. et al., 2019) and an updated secondary organic aerosol scheme in the troposphere (Tilmes et al., 2019). It further uses a modal aerosol scheme (MAM4) for both troposphere and stratosphere (Liu et al., 2016) and prognostic sulfur injection to simulate eruptive volcanoes during the historical period (Mills et al., 2016, 2017). The atmospheric model is coupled to the other components in CESM2. The Parallel Ocean Program version 2 (POP2) (Smith et al., 2010; Danabasoglu et al., 2012) includes several improvements compared to earlier versions, including ocean biogeochemistry represented by the Marine

Biogeochemistry Library (MARBL), which incorporates the Biogeochemical Elemental Cycle (BEC) ocean biogeochemistry-ecosystem model (e.g., Moore et al., 2014; Harrison et al., 2018) and the NOAA WaveWatch-III ocean surface wave prediction model (Tolman, 2009). Additional components are the sea-ice model CICE version 5.1.2 (CICE5) (Hunke et al., 2015) and the Community Ice Sheet Model version 2.1 (CISM2.1), (Lipscomb et al., 2019). The Community Land Model version 5 (CLM5) also includes various updates, including interactive crops and irrigation for the land (Lawrence et al., 2019), and the Model for

Scale Adaptive River Transport (MOSART).

CESM2 and WACCM6 have contributed to the Coupled Model Intercomparison Project phase 6 (CMIP6) (Eyring et al., 2016). As part of CMIP6, WACCM6 performed the DECK simulations, as well as the historical simulations, which reproduced the observed surface temperature trend within the expected variability (Gettelman et al., 2019).

### 2.2 Description of the testbed experiments

For the testbed baseline scenarios, the SSP5-85 and SSP5-34 OS CMIP6 scenarios are used. SSP5-85 starts in 2015 from a historical simulation and ends in 2100 (O'Neill et al., 2016). Anthropogenic, biomass burning, ocean, soil, and volcanic emissions are prescribed, as well as surface concentrations of greenhouse gases and land surface values, using the corresponding





scenarios (Meinshausen et al., 2017), while biogenic emissions are interactively calculated. SSP5-34-OS concentrations follow the SSP5-85 scenario until 2040 and then drastically change path. SSP5-85 $CO_2$ concentrations continuously increase after
2040 until the end of the 21st century, reaching up to 1100 ppm, and methane ($CH_4$) concentrations increase until 2070 and slowly decline thereafter (Fig. 1, bottom panel). For SSP5-34-OS, strong mitigation efforts are set in place after 2040, as well as the inclusion of negative emissions. Nevertheless, $CO_2$ concentrations still grow until about 2060 reaching $\approx$ 550 ppm and then slowly decline by the end of the 21st century, reaching $\approx$ 500 ppm based on WACCM6 simulations. $CH_4$ concentrations drop relatively quickly after 2040, due to its much shorter lifetime than $CO_2$, reaching values of 1 ppb by the end of the 21st
century. This is assuming a drastic phase-out of any anthropogenic production of CH4 after 2040.

Two climate intervention experiments are designed to use the same prescribed greenhouse gas concentrations, emissions, and land surface values as the baseline SSP5-34-OS scenario. The experiments are designed to maintain global mean near surface temperatures around $1.5^oC$ and $2.0^oC$ warming compared to 1850-1900 levels, respectively, and are called "Geo SSP5-34-OS 1.5" and "Geo SSP5-34-OS 2.0". The start of each climate intervention experiment is defined by the time that the baseline
simulation has reached near-surface global-mean temperature of $1.5^oC$ and $2.0^oC$ above pre-industrial, considering a ten-year running mean (in WACCM6 this is around 2020 for $1.5^oC$ and 2034 for $2^oC$ for the first ensemble member).

Besides global mean surface temperature targets we require two more surface temperature measures in the proposed experiments, namely interhemispheric temperature gradients and equator to pole temperature targets, as described in Kravitz et al. (2016) and MacMartin et al. (2017). These additional temperature targets are defined based on the period when global mean
surface temperatures have reached the specific climate goals, see above. Sulfur dioxide injections into the stratosphere are performed at 4 locations, at $15^oN$, $15^oS$, $30^oN$, and $30^oS$, following the approach described in Kravitz et al. (2017) and Tilmes et al. (2018). A feedback algorithm has been applied that was developed by MacMartin et al. (2017), based on an earlier version of WACCM (Mills et al., 2017). The injection rate each year is computed based on an initial guess (a "feed-forward") that is corrected based on the actual temperature history (the "feedback"). The feed-forward function helps the controller more easily
to reach the goals. This algorithm has been adopted in the WACCM6 without any changes, despite using a slightly different scenario in WACCM6 (using SSP5-85) compared to GLENS (using RCP8.5). For the OS simulations, the same feedback algorithm was applied, but with changes to the feed-forward function to account for the different temperature evolution in the baseline simulation.

In this study, only one realization of the proposed testbed experiments has been used due to computational limitation and one
additional ensemble is in progress. Since the SSP5-85 scenario is identical to SSP5-34-OS until 2040, we started the SSP5-34-OS in 2040 from the SSP5-85 scenario. WACCM6 near surface temperatures reached around $1.3^oC$ warming compared to the 1850-1900 average by 2015 and $1.5^oC$ around 2020 using the first WACCM6 ensemble member from the historical simulation (Fig. 1, top panel). The global mean surface warming reaches $6.3^oC$ by 2100. The SSP5-34-OS global mean surface temperature reaches up to $3^oC$ above the 1850-1900 temperature by 2060, aligned with the maximum peak in $CO_2$ concentrations.
Temperatures slightly decline by the end of the century to about $2.5^oC$ above pre-industrial. Global near surface temperature targets were reached in the two SAG model experiments within about $0.2^oC$ (Fig. 1, top panel, green and orange lines). In addition to the two proposed SAG model simulations, we also performed a third climate intervention experiment that uses





SSP5-85 as the baseline scenario, while applying sulfur injections to keep near surface temperature levels at $1.5^oC$ targets, called "Geo SSP5-85 1.5". Note that this scenario is identical to the "Geo SSP5-34-OS 1.5" experiment between 2015 and
2040 (Fig. 1, top panel, purple line).

Comparing the outcomes of Geo SSP5-85 1.5 with the Geo SSP5-34-OS 1.5 experiment, allows us to explore the differences of the impact of SAG using a high forcing greenhouse gas scenario vs. the overshoot scenario after 2040. Geo SSP5-85 1.5 can also be compared to the results in GLENS, since it uses the same setup with a similar baseline simulation but different model versions. GLENS was performed with an earlier WACCM version 5.4 (Mills et al., 2017). GLENS simulations include
a 3-member ensemble of the future baseline simulation starting in 2010, following the RCP8.5 pathway, called "RCP8.5" in the following. GLENS SAG simulations reach the same surface temperature targets of $1.5^oC$ and are called "Geo RCP8.5 1.5" in the following (see Table 1).

## 3  Sulfur injection rates, burden, and efficiency

The feedback algorithem calculates the required injection amount per injection location after each year of the simulation, based
on the surface temperature deviations from the target temperatures. For all of the cases, a larger fraction of the injection was placed into the Southern Hemisphere (SH) (Fig. 2). For Geo SSP5-85 1.5, the injections were mainly placed at $30^oN$ and $30^oS$, with a slightly smaller amount in the Northern Hemisphere (NH). Only half of the amount that was used at $30^oS$ was required at $15^oS$ and almost no injection was required at $15^oN$ to achieve the predefined temperature goals. For the Geo SSP5-34-OS 1.5 and Geo SSP5-34-OS 2.0 experiments, most injections were placed at $30^oN$, $30^oS$, and $15^oS$. After 2080 for Geo SSP5-
34-OS 1.5 (2070 for Geo SSP5-34-OS 2.0) only injections in the SH were needed, and injections at $15^oS$ dominated. As a result, the sulfate loading is significantly larger over the SH than the NH. This is in contrast to what has been simulated in Geo RCP8.5 1.5 (GLENS), where more injections were required in the NH in order to achieve the same temperature targets (Tilmes et al., 2018). An in depth investigation is needed in future studies to understand the differences using the two different CESM model versions, however, differences may be in part connected to differences in the ocean response, described in Section 4,
and potentially as a result of differences in anthropogenic sulfur emissions.

Differences between the 3 SAG experiments and the Geo RCP85 1.5 also arise in terms of injection amount and the resulting aerosol burden (Table 1 and Fig. 3). The maximum injection amount in Geo SSP5-85 1.5 is 48 $TgSO_2$ per year with a total burden reaching up to 25 TgS. This results in an accumulated injection amount of 1710 $TgSO_2$ by the end of the century (Table 1). In contrast, Geo RCP85 1.5 required a larger injection with an accumulated injection amount of 2056 $TgSO_2$ and
a corresponding burden of 28 TgS. The correlation between sulfur burden and injection rate is similar between Geo RCP85 1.5 and Geo SSP5-85 1.5 (Fig.3, bottom panel), which concludes that production, transport and removal processes in the two WACCM versions are similar. The reason for the slightly smaller required injection amount in Geo SSP5-85 1.5 compared to Geo RCP8.5 1.5 could be differences in the baseline scenarios, which specify a larger sulfate burden in the tropospheric in SSP5-85 compared to RCP8.5 (not shown).




The two testbed SAG experiments that are based on the OS scenario show much reduced accumulated $SO_2$ injections compared to the high forcing scenarios, with 605 Tg $SO_2$ for the 1.5$^o$C temperature target and 305 Tg $SO_2$ for the 2.0$^o$C temperature target. For Geo SSP5-34-OS 1.5, the total annual injection peaks between 2050 and 2070 at 10-12 Tg $SO_2$, an amount comparable to the observed global sulfate perturbation from the 1991 eruption of Mt. Pinatubo (Baran and Foot, 1994; Dhomse et al., 2014; Mills et al., 2016). For Geo SSP5-34-OS 2.0, injections peak around 2050, reaching about 9 Tg$SO_2$,

and falling off after that towards around 1 Tg$SO_2$ injections per year by the end of the century. In particular for the OS cases, there were periods in which the near surface temperatures were slightly cooler than the target temperature (e.g. between 2050 and 2070 for Geo SSP5-34-OS 2.0). This was due to shortcomings in the feed-forward component of the controller setup for the SSP5-34-OS 2.0; in particular, the feed-forward was estimated based only on the instantaneous cooling required and did not adequately take into account the "memory" in both the aerosol concentrations and the resulting temperature response.

The feed-forward component thus overestimated the amount of $SO_2$ injection required once aggressive mitigation began; this was eventually successfully corrected by the feedback. Both experiments that are based on the OS baseline scenario show reduced values of sulfate burden vs. sulfur dioxide injections (Fig. 3, bottom panel) for the years when $SO_2$ injections have been declining because of the prevalent sulfate burden from previous years.

## 4   Impacts of stratospheric aerosol geoengineering

### 4.1   Surface air temperature changes

The design of the proposed testbed experiments allows us to assess the effects of SAG, while surface air temperatures are maintained at specific targets, here 1.5$^o$C and 2.0$^o$C above pre-industrial levels. Since 1.5$^o$C of warming, the more desired temperature target defined by the IPCC1.5 report, is reached around 2020 (2015–2025) in the WACCM6 SSP5-85 simulation, we use this period as the control period for our analysis. Results in Figs. 4 and 5 are therefore illustrated in reference to

2015–2025 control values based on SSP5-85. The evolution of global mean surface air temperatures in the different experiments has been described above. Here, we discuss the surface air temperature evolution in NH and SH, in order to illustrate interhemispheric temperature differences, Fig. 4, solid and dotted lines, respectively, for the different experiments.

The two baseline simulations (SSP5-85 and SSP5-34-OS) show an increase in deviations of hemispheric surface air temperatures from the global mean temperature. While in SSP5-85, interhemispheric temperature differences continue to increase

towards the end of the 21st century with stronger temperature trends in the NH compared to the SH, interhemispheric temperature differences in SSP5-34-OS reverse around 2070. This results in very small temperature trends in the SH after 2070 and decreasing temperatures in the NH. In WACCM6, NH temperatures are strongly impacted by the so called "warming hole" in the North Atlantic, which describes a local cooling that counters increasing temperatures from increasing greenhouse gases (Fig. 5, top panel). The cooling of surface air temperatures above the North Atlantic is similar in magnitude for both SSP5-85

and SSP5-34-OS, likely a result of a fairly similar slowdown of the AMOC, as discussed in Section 4.2. On the other hand, the warming in the NH due to increasing greenhouse gases is much larger in SSP5-85 than in SSP5-34-OS, resulting in the differences in North-to-South temperatures between the two baseline scenarios.





Applying the feedback algorithm to SSP5-85 and SSP5-34-OS results in a removal of the interhemispheric gradient in addition to maintaining global mean surface air temperatures. Only the last 15 years (2085–2100) of the Geo SSP5-34-OS

2.0 experiment produces somewhat larger warming in the SH than in the NH (Fig. 4, left panels). Zonal mean surface air temperature changes from the different experiments are illustrated for two different periods in Fig. 4, middle and bottom panels on the left. For the baseline simulations, temperatures in high latitudes are higher than in mid and low latitudes, as expected, leading to much larger warming than the global mean. Effects of the warming hole (cooling) in the North Atlantic are visible (Fig. 5), particularly for the SSP5-34-OS scenario towards the end of the 21st century. SAG applications show a significant

reduction in the warming of the polar regions, with very little difference between pole and equator in all the sulfur injection experiments. Only a slight warming up to $1^oC$ occurs in the SH polar region by the end of the 21st century. The continuous cooling in the North Atlantic (Fig. 5, middle panels) is compensated by a warming over Northwest Europe. Temperature goals are therefore reached equally well in all the sulfur injection experiments, using different baseline scenarios. The Geo SSP5-34-OS 2.0 is slightly cooler in the NH and shows a slight warming in the SH compared to the temperature target. This experiment

is designed to be $0.5^oC$ warmer than the other two SAG experiments. Therefore, independent of reaching $1.5^oC$ and $2.0^oC$ temperature targets, the feedback approach is able to maintain zonally averaged surface air temperatures at most latitudes.

## 4.2  Atlantic Meridional Overturning Circulation changes

Sea surface temperature (SST) anomalies are significantly reduced by SAG in all scenarios. Simulated present day (2015–2024) SST is already significantly warmer than pre-industrial (PI) across the tropics, subtropics, and into the Southern Ocean, with

anomalies between $0.5^oC$ and $1.5^oC$, reaching $2^oC$ in the equatorial Pacific (Fig. A1). On top of this, in the 2060s simulated SST is significantly warmer than 2015–2024, with broad regions reaching anomalies above $2^oC$ in the SSP5-85 case and $1.5^oC$ in the SSP5-34-OS case; the exception is the warming hole in the North Atlantic (Drijfhout et al., 2012), which is significantly and persistently cooler by $1–2^oC$ from both PI and present day SST by 2070, even in SSP5-85 (Fig. A2). SST anomalies are largely reduced in all geoengineering protocols implemented in this study especially in the $1.5^oC$ cases. The exception is the

warming hole, which remains persistently cool in all scenarios, regions of the Arctic which remain slightly warm, and small regions of warming in the Indian sector of the Southern Ocean. Regions of persistently warm anomalies remain in the $2.0^oC$ case, including much of the eastern Indian Ocean, the equatorial and western North Atlantic, and east of Japan in the North Pacific.

The apparent warming hole in all of the simulations is very likely related to changes in the AMOC (Fig. 6). The baseline

scenarios SSP5-85 and SSP5-34-OS show a very similar decline until the last 2 decades of the simulation, with a maximum decline of more than 50% by the end of the century. Both SAG scenarios that target the $1.5^oC$ temperatures show only a relatively small decline from 2020 values (approx 25%), with the largest reduction during the last 20 years of the simulation. The Geo SSP5-34-OS 2.0 produces a stronger decline closer to 40% and therefore closer to the SSP5-34-OS baseline scenario.

In comparison, Geo RCP85 1.5 (GLENS) simulations do not show the relative cooling in the North Atlantic (Fasullo et al.,

2018). The earlier version of the model shows a slowing of the AMOC for the RCP8.5 scenario similar to the WACCM6 CMIP6 SSP5-85 simulation, which is however much smaller. Danabasoglu et al. (2019) found that the maximum AMOC strength in





CESM2 is stronger than in CESM1. The differences in AMOC between CESM1 and CESM2 reflect differences in water mass properties that are ascribed (partly) to surface flux differences, as the ocean model component in both model versions handles the dense-water overflows through the Denmark Strait and the Faroe Bank Channel in the same way. Applying SAG resulted

in an acceleration of the AMOC in GLENS (Fig. 6, grey shaded area), which is not the case in any of the WACCM6 SAG simulations. In these simulations the AMOC is still declining, even though less severely than in the SSP5-8.5 simulation. Responses of AMOC and therefore effects on surface air temperatures seem to be largely model version dependent.

### 4.3   Zonal mean precipitation changes

Global mean precipitation is changing compared to the 2015–2025 control, even though global surface air temperatures are

maintained using SAG, as expected based on various earlier studies. Similarly to what has been found in Tilmes et al. (2016), and Jones et al. (2018), precipitation is increasing for the baseline scenarios, while applications of a low forcing scenario result in close to present day global precipitation values. In WACCM6, precipitation is declining the most compared to 2020 values in Geo SSP5-85 1.5, with increasing reductions towards the end of the century, aligned with the increasing amount of sulfur dioxide injections, which is very similar to what has been found in GLENS (Fasullo et al., 2018, e.g.,). However, the SAG

experiment based on the OS pathway and aiming for the $1.5^{o}C$ target, results in a much smaller global mean precipitation change. Furthermore, Geo SSP5-34-OS 2.0 shows a slight increase in global mean precipitation with increasing values after 2070.

    Large scale precipitation changes from the control are shown in the zonal mean precipitation anomalies (Fig. 4, middle and bottom panels on the right). Both baseline simulations (SSP5-85 and SSP5-34-OS) show increasing precipitation in tropics and

mid to high latitudes between 2060–69. While this trend is continuing in SSP5-85, the SSP5-34-OS shows a reduction in the precipitation changes compared to control, as a result of reduced warming in this scenario by the end of the 21st century. A shift in tropical precipitation towards the SH (and therefore a shift in the Inter-tropical Convergence Zone (ITCZ)) occurs and is most pronounced in the SSP5-85, with increasing intensity towards the end of the 21st century in both baseline scenarios. Despite the reduction in greenhouse gases and surface temperature relative to SSP5-85, impacts on tropical precipitation using

the overshoot scenario are still large and may result in large regional impacts. SAG applications successfully reduce increasing precipitation and shifts in tropical precipitation in 2060–2069, with slight reductions in precipitation in the SH subtropics. Some larger differences occur by the end of the 21st century, where reductions in precipitation are most pronounced if using the SSP5-85 baseline scenario. Also, we identify a shift in tropical precipitation for Geo SSP5-34-OS 2.0, which is likely a result of the occurrence of an interhemispheric temperature gradient in this scenario by the end of the 21st century. More

detailed investigations have to be performed in future studies, as well as in a multi-model comparison context. Precipitation changes are therefore strongly dependent on the amount and strategy of SAG application.

### 4.4   Land Primary Productivity

Net primary productivity (NPP) over land is the difference between gross primary productivity (GPP) and plant respiration (Cramer et al., 1999), and it is a key component in the terrestrial carbon cycle. NPP is sensitive to climate changes, including





temperature, precipitation, soil moisture and photosynthetically active radiation. As shown in previous analysis (Cheng et al., 2019), relative to the baseline, SAG would reduce temperature, change precipitation and evaporation, which would potentially change soil moisture, and reduce the total incoming solar radiation. Therefore, terrestrial NPP is influenced by SAG.

Fig. 7 shows the accumulated annual land NPP in different baselines and SAG scenarios. Here NPP shows strong dependency on $CO_2$ concentration, consistent with previous studies (Govindasamy, 2002; Kravitz et al., 2013; Glienke et al., 2015). In

CLM5, $CO_2$ concentration is one of the factors to determine the stomatal resistance and photosynthesis rate (Lawrence 2019). With higher $CO_2$ concentration in SSP5-85 and Geo SSP5-85 1.5, plants tend to have less stomatal conductance which makes them more resistant to water stress, and to have higher photosynthesis rate. Therefore, land NPP in those two scenarios increases constantly through the whole simulation period. With mitigation and carbon dioxide removal strategy, $CO_2$ concentration under SSP5-34-OS and the related SAG scenarios reaches a maximum around 2060, and then reduces slowly. In general, land NPP

in our simulations follow the change of $CO_2$ concentrations in the baseline. Temperature reduction or other climate changes from SAG show mild impact on land accumulated NPP. However, comparison between baseline and SAG indicates regional different responses of land NPP to SAG climate changes.

Figure 8 shows NPP anomalies between the three SAG scenarios and their baseline during 2060–2069. There are similar patterns in the maps with SAG, where land NPP increases over tropical and midlatitude regions, while it decreases over high

latitude and high altitude areas. The temperature reduction from SAG plays an important role in this pattern citep[]Kravitz2013. Lower leaf temperature over tropical and midlatitude regions enhances stomatal conductance and hence promotes the carbon gain, while over high latitude and high altitude regions, the cooling is not optimal for plant growth. The magnitude of changes depend on both baseline and the temperature target. With a larger temperature difference between the baseline and the SAG, the NPP changes are bigger. As shown in Fig. 7, NPP changes are the largest between SSP5-8.5 and Geo SSP5-8.5 1.5.

## 4.5 Ocean ecosystem impacts

Warming has large impacts on ocean ecosystems and fisheries, both directly through ocean temperature impacts on physiological processes, and indirectly through warming-induced changes in ocean physics. Increases in ocean temperature elevate respiration rates for endothermic (cold-blooded) animals, including zooplankton and fish, decreasing body size and limiting energy transfer to commercial fishery species and large marine vertebrates (Heneghan et al., 2019; Lotze et al., 2019). In con-

trast, warming ocean temperatures may stimulate NPP by phytoplankton, marine primary producers that make up the base of the marine food-web, assuming no other changing conditions (Eppley, 1972; Krumhardt et al., 2017). Additionally, however, warming drives changes in ocean stratification, currents and other physical mechanisms (clouds, sea ice, river flow) that affect nutrient delivery processes and available light (Laufkötter et al., 2015; Lauvset et al., 2017; Harrison et al., 2018). For example, warming induced stratification increases in pelagic ecosystems may reduce the amount of nutrients supplied to the photic zone,

decreasing marine NPP, indirectly impacting higher trophic levels. Combined together, net responses of marine ecosystems to climate perturbations are dependent on local physical and biogeochemical conditions, leading to diverse ecosystem responses in different regions (Bopp et al., 2013; Krumhardt et al., 2017; Lauvset et al., 2017). Globally integrated, these processes are predicted to cause a net decrease of globally integrated oceanic biological production in future climate scenarios (Krumhardt





et al., 2017), with a projected 5% decline in fisheries production for every degree of surface temperature warming (Lotze et al.,
2019). Here we investigate to what degree solar radiation management mitigates the primary drivers of marine ecosystem
disruption, sea surface temperature and net primary productivity.

Anomalies outside historical climate variability are one indication of ocean conditions that ecosystems are not adapted to, and
thus expected to cause disruption to fisheries and natural ecosystems (Bopp et al., 2013; Heneghan et al., 2019). Accordingly,
significance of SST (Fig. A2) and NPP (Fig. 9) anomalies was determined by using the standard deviation ($\sigma$) in each model
grid cell of the yearly means from the 499 year pre-industrial control run. An anomaly was considered significant when it was
greater than 1.96 $\sigma$ (95% confidence interval).

Oceanic NPP, the rate of photosynthetic carbon fixation by marine phytoplankton (Krumhardt et al., 2017; Harrison et al.,
2018), represents the base of marine food web, supporting fisheries and natural ecosystems and driving the biological carbon
pump that removes $CO_2$ from the atmosphere (e.g., Sarmiento and Gruber, 2006; Harrison et al., 2018). Similar to previous
Earth system model simulations, anomalies of NPP in future climate are highly variable in space, and feature both strong
positive and negative anomalies (Fig. 9), driven by different mechanisms in different biomes (Bopp et al., 2013; Krumhardt
et al., 2017). In contrast to SST, simulated NPP is not significantly different in 2015–2024 relative to PI over much of the
global ocean, with the exception of increased NPP at the poles, where both declining ice and warming temperatures increase
production, and a narrow strip at the subtropical-subpolar boundary in the Southern Hemisphere (Fig. A1); these anomalies
get stronger by 2070 (Fig. 9). Additionally, the North Atlantic warming hole is associated with NPP declines of 30–40%,
likely caused by changes in nutrient supply. All anomalies are substantially mitigated by SAG, with positive NPP anomalies
relative to present disappearing over much of polar oceans, and NPP reductions in the North Atlantic decreasing from 30–40%
(baseline cases) to 20–30% in the 1.5°C SAG cases. Thus, SAG could reduce negative impacts of climate change on marine
ecosystems in the North Atlantic, an important region for fisheries. It is important to note, however, that the ocean ecosystem
model in CESM2 does not account for the effects of ocean acidification on marine phytoplankton, which could impact, for
example, calcifying phytoplankton (Krumhardt et al., 2019) or diatoms (Bach et al., 2019; Petrou et al., 2019).

### 4.6 Ice sheet mass balance

The mass balance (MB) of (grounded) ice sheets, which determines their contribution to sea level rise, is made up from two
components: the surface mass balance (SMB; representing snowfall and surface melt), and solid ice discharge (D) across the
grounding line (Lenaerts et al., 2019)

MB = SMB - D

As D is controlled by ice flow speed and ice thickness, and responds relatively slowly to external forcing, it is challenging to
detect an impact from SAG on ice discharge within a single century. Moreover, default CESM2 and therefore WACCM6 does
not explicitly represent D, as it requires a dynamic ice sheet model coupled to the ESM, a feature that is currently only available
in dedicated CESM2 experiments (Lipscomb et al., 2019). SMB, on the other hand, is explicitly represented in CESM2, as
it is primarily driven by atmospheric and surface processes, in particular snowfall and surface melting, and therefore has a
much shorter response time. In addition, while ice sheet SMB exhibits large interannual variations, it also is observed to show





a discernible trend on ice sheets in both hemispheres (Lenaerts et al., 2019). The observed Greenland Ice Sheet mass loss and associated sea level rise is primarily driven by a declining SMB (van den Broeke et al., 2016), and will very likely continue

to do so in the future (Aschwanden et al., 2019). A common tipping point for the Greenland Ice Sheet (GrIS) is assumed to SMB = 0, when the ice sheet no longer has a mechanism to gain mass; this threshold is likely already reached this century in higher-emission scenarios (Pattyn et al., 2018). In contrast, the Antarctic Ice Sheet SMB has increased throughout the past century (Medley and Thomas, 2019), potentially acting to mitigate Antarctic mass loss through increasing D. While we are not able to identify the impact of SAG on Antarctic D, recent studies indicate that the Antarctic Ice Sheet will likely become

unstable (leading to a sharp increase in D) when we increase global mean temperature by above 2$^o$C (Pattyn et al., 2018).

In Figure 10, a general decrease in GrIS SMB is seen in all simulations compared to the historical period, but most notably in the high-warming scenarios SSP5-85 and SSP5-34-OS. This decrease is driven by increased surface runoff (Fig. A3) which is only partly offset by increased snowfall (not shown). SAG is effective in stabilizing runoff and therefore SMB in all three simulations, albeit there still is a distinct departure from late 20th century values. Although this is good news for the stability

of the GrIS, and the tipping point SMB=0 is only reached in SSP5-85, it does not guarantee the GrIS existence in the long run since we do not resolve discharge. Moreover, the SMB-elevation feedback is not explicitly modeled, which starts to play a dominant role on millennial time scales (Pattyn et al., 2018). Based on these results, we deem it unlikely that large freshwater fluxes will originate from the GrIS by surface processes alone in all 3 geoengineering scenarios.

Throughout most of the 21st century, the response of AIS SMB is similar in all SAG simulations (Fig. 10, right panel).

Again, both a stabilization and a marked departure are seen from 1960–1999 values, suggesting that the response time of SMB to warming is in the order of years-decades. In contrast to the GrIS, SMB increases during the 21st century, which is explained by the dominant role of precipitation on the AIS, whereas surface runoff remains a comparatively small mass flux (Fig. A3). Interestingly, simulations Geo SSP5-34-OS 1.5 and SSP5-85 1.5 depart from one another during the second half of the 21 century. We attribute this difference to the different aerosol loading in the two simulations, which impacts the formation of

precipitation.

### 4.7  Evolution of the Antarctic ozone hole

The annually recurring ozone hole over Antarctica that began around 1980 is a result of enhanced CFCs and other halogen reservoirs in the stratosphere, the so-called ozone destroying substances (ODS), that mostly accumulated before the 1990s. Due to their very long lifetime of some CFCs over 100 years, the burden of ODS peaked around the year 1990 and is now

slowly declining. The Antarctic ozone hole is expected to recover back to 1980 values in 2060 (WMO, 2018). However, changes in surface climate due to anthropogenic climate change are projected to accelerate the Brewer-Dobson Circulation in the stratosphere and with that transport more ozone into high latitudes and increase ozone with time, which can lead to a "super recovery" of ozone. The larger the forcing scenario, the larger is this effect, which would potentially slightly speed up the recovery of the ozone hole. RCP8.5 simulations as part of GLENS show the recovery of the Antarctic ozone in October by

around 2060 compared to 1980 total column ozone values (Fig. 11, top panel) and an increase of column ozone up to 30 DU by the end of the century. The same behavior is also shown in WACCM6 following SSP5-85. Overshoot scenarios also show





a recovery to 1980 values, which stays at or slightly below that value for the rest of the simulations, as a result of the reduced climate change effect on ozone (Fig. 11, middel panel).

The increasing aerosol burden in the stratosphere in the SAG modeling experiments has significant effects on stratospheric
chemistry and transport (e.g., Tilmes et al., 2009; Ferraro et al., 2014; Richter et al., 2018). The absorption of radiation by sulfate aerosols heats the lower tropical stratosphere. The amount of heating is proportional to the sulfur injection amount and results in a drop of the tropopause altitude and an increase in tropopause temperatures (Tilmes et al., 2017). These changes, in addition to the cooling of the surface and the troposphere, influence the strength of the sub-tropical and polar jets and therefore transport of stratospheric airmasses. In addition, stratospheric aerosols increase the aerosol surface area important for
heterogeneous reactions. This leads to an enhanced activation of chlorine and therefore increased ozone depletion. The effect of SAG was estimated to delay the recovery of the ozone hole by at least 40 years (Tilmes et al., 2008).

Both GLENS and WACCM6 simulations show a drop in Antarctic column ozone at the start of the SAG application between 2020 and 2030 of up to 70 DU and then an increasing trend, similar to the case without SAG application. Antarctic column ozone has not fully recovered in Geo SSP5-85 1.5 by the end of the century. On the other hand, the SAG scenarios Geo SSP5-
34-OS 1.5 and 2.0 show a faster recovery of the ozone hole than Geo SSP5-85 1.5, which is reached around 2080. The reduced forcing scenario does require less sulfur injections to reach the temperature targets, which results in a smaller stratospheric aerosol burden. Therefore, less ozone depletion is expected and the delay of the recovery of the ozone hole would be shortened to 20–30 years. For SSP5-34-OS 2.0, the later start of SAG application leads to a weaker reduction of column ozone of around 45 DU compared to the drop in column ozone of 70 DU if SAG would be started in 2020.

## 5   Discussions and conclusions

This paper describes a new testbed GeoMIP scenario for CMIP6, which aims to keep global warming to less than $1.5^o$C and $2.0^o$C above pre-industrial. Two different baseline scenarios are used, the high greenhouse gas emission scenario SSP5-85 and the overshoot scenario SSP5-34-OS, which follows the SSP5-85 future pathway until 2040, and then drastically increases decarbonization afterwards. We describe three different stratospheric aerosol geoengineering simulations. The first uses the
SSP5-85 baseline scenario and performs stratospheric sulfur injections to keep surface air temperatures at the $1.5^o$C temperature target, while starting sulfur injections in year 2021 (Geo SSP5-85 1.5) using WACCM6. The second experiment uses the SSP5-34-OS baseline scenario and is also designed to keep surface air temperatures at the $1.5^o$C temperature target (Geo SSP5-34-OS 1.5). Both Geo SSP5-85 1.5 and Geo SSP5-34-OS 1.5 are identical until the year 2040. The third experiment uses the SSP5-34-OS baseline scenario, but this time is designed to keep surface air temperatures at the $2.0°$C temperature target,
starting in 2034 (Geo SSP5-34-OS 1.5) in WACCM6.

The purpose of the different simulations is to explore the range of outcomes of SAG dependent on the amount of SAG injections, the background $CO_2$ concentrations, and the target surface air temperatures. The overshoot baseline scenario is used because it provides the opportunity to study the effects of limiting SAG applications in duration and amount. The proposed SAG testbed simulation starts in WACCM6 around 2020 or 2034 (which could be somewhat different in other models), a





period when the overshoot scenario is still on an increasing greenhouse gas emission path. We do not recommend to consider applications of SAG before large scale mitigation measures have been adopted and therefore consider this scenario not to be policy relevant. More realistic and policy relevant scenarios need to be designed in the future that include earlier actions on mitigation, more realistic implementation of potential negative emissions and assumed surface emissions. However, we decided to use it, because it is the best scenario that has been designed and is available to the CMIP6 modeling community to be used

for CMIP6.

Proposed SAG modeling experiments need to be designed in a strategic manner in order to achieve the least amount of side effects. For example, Kravitz et al. (2017) have shown several improvements in using the feedback controller to achieve the three temperature targets. Surface air temperature targets are relevant for reducing effects including extreme temperatures, heatwaves, and sea ice melting. The better outcomes can be reached using the feedback algorithm to keep temperature gradients

between North and South and between the Equator and the poles from changing.

Applications of the three temperature targets in WACCM6 result in small differences in the amount of warming in high latitudes between a 1.5$^o$C and a 2$^o$C temperature target. Therefore, differences that were described in the IPCC1.5 report (Masson-Delmotte et al., 2018) between reaching 1.5$^o$C or 2$^o$C target may be different if they are reached with SAG or with emission reductions only, and have to be investigated further. On the other hand, global precipitation changes depend on the

amount of sulfur injections, resulting in a stronger reduction with increasing application. Precipitation changes and shifts in the ITCZ occur in both baseline scenarios, and in the OS case by the end of the century. This is likely a result of changes in the distribution of tropospheric aerosols. SAG using the feedback algorithm helps to reduce these shifts, whereby reduction in precipitation is strongest with higher injection amounts and to a lesser amount depends on the temperature targets. For the testbed experiment, it would be most desirable to implement this feedback controller in other models to be able to reach similar

temperature targets.

The impacts of SAG need to be explored within the entire space between scenarios and societal and ecological relevant impacts to holistically assess and improve SAG applications. Here we provide examples of how such an assessment could be established, considering different types of scenarios, e.g., high greenhouse gas scenarios, low GHG scenarios, high vs. low SAG, and differences in temperature targets. All of these matter for different impact variables in a different manner. There

are many different variables that need to be investigated. This paper explores only a few of those variables and illustrate their dependency on impacts based on temperature targets, amount of sulfur burden, and the baseline simulations (Table 2). Furthermore, differences in injection amounts will impact costs of the implementation and need to be taken into account, but have not been investigated here.

Changes in AMOC, that are coupled to the surface temperatures, lead to a significant warming hole in WACCM6 with

consequences for ocean temperatures, reducing NPP in the ocean in the North Atlantic. The reduced slowing of the AMOC with SAG would decrease some impacts on marine ecosystems. However, SAG will not mitigate other ecosystem stressors, like ocean acidification, which depend on the baseline scenario. Land NPP is also strongly dependent on the $CO_2$ content of the atmosphere and therefore on the baseline simulations but not so much on the temperature target. On the other hand, mean ice sheet surface mass balance is strongly dependent on the surface temperature target and has only a small direct dependence





on the amount of SAG application or the baseline simulations. Finally, the Antarctic ozone hole is expected to recover around 2060 without SAG, but cannot fully recover by the end of the century if SAG would be applied to the SSP5-85 baseline scenario to reach $1.5^oC$. Using the OS scenario, ozone super-recovery is reduced and SAG applications would delay the recovery by approx. 20–30 years until around 2080, with a slightly early recovery if the $2^oC$ target would be used.

In summary, future changes in different quantities that are important for societal and ecological impacts depend on very
different measures, including the amount of SAG application, temperature target and baseline simulation. A comprehensive assessment is required that holistically considers benefits and side effects of climate intervention strategies. Multi-model experiments are needed to identify the range of outcomes and uncertainties.

*Data availability.* Previous and current CESM versions are freely available (www.cesm.ucar.edu:/models/cesm2). The CESM2 data ana-
lyzed in this manuscript have been contributed to CMIP6 and are freely available at the Earth System Grid Federation (ESGF; https://esgf-node.llnl.gov/search/cmip6/) or from the NCAR Digital Asset Services Hub (DASH; https://data.ucar.edu) or from the links provided from the CESM website (www.cesm.ucar.edu).

*Author contributions.* As the first author, Simone Tilmes organized and wrote a large portion of the paper, produced all the simulations, and performed the analysis presented in about half of the figures. Douglas E. MacMartin was instrumental in properly setting up the feedback algorithm, and contributed to the interpretation of results and the writing of the paper. Jan T. Lenaerts and Leo van Kampenhout analyzed
land ice mass balance and produced related figures and text. Laura Muntjewerf produced AMOC figures and the corresponding text. Lili Xia analyzed land primary productivity and produced related figures and text. Cheryl S. Harrison and Kristen M. Krumhardt analyzed sea surface temperature and ocean net primary productivity, producing related figures and text. Michael J. Mills, Ben Kravitz, and Alan Robock contributed to the writing of the paper and the overall framing.

*Competing interests.* There is are competing interests at present

*Acknowledgements.* We thank Anne K. Smith and Douglas E. Kinnison for helpful comments and suggestions. Support for Ben Kravitz was provided in part by the National Science Foundation through agreement CBET-1931641, the Indiana University Environmental Resilience Institute, and the Prepared for Environmental Change Grand Challenge initiative. The Pacific Northwest National Laboratory is operated for the US Department of Energy by Battelle Memorial Institute under contract DE-AC05-76RL01830. Alan Robock was supported by NSF grant AGS-1617844. The CESM project is supported primarily by the National Science Foundation. This material is based upon work
supported by the National Center for Atmospheric Research, which is a major facility sponsored by the NSF under Cooperative Agreement No. 1852977. Computing and data storage resources, including the Cheyenne supercomputer (doi:10.5065/D6RX99HX), were provided by the Computational and Information Systems Laboratory (CISL) at NCAR. Part of this work was carried out under the program of the





Netherlands Earth System Science Centre (NESSC), financially supported by the Ministry of Education, Culture and Science (OCW)".
(Grantnr. 024.002.001).



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





**Table 1.** Overview of model simulations

| Experiment | Emission and Concentration Pathway | Accumulated $SO_2$ injection | Max Surface Air Temperature from PI |
|---|---|---|---|
| SSP5-85 | SSP5-85 | 0. | 6.3 |
| SSP5-34-OS | SSP5-34-OS | 0. | 3.0 |
| RCP-85 | RCP-85 | 0. | 6.5 |
| Geo SSP5-85 1.5 | SSP5-85 | 1710 Tg $SO_2$ | 1.5C |
| Geo RCP-85 1.5 | RCP-85 | 2056 Tg $SO_2$ | 1.5C |
| Geo SSP5-34-OS 1.5 | SSP5-34-OS | 605 Tg $SO_2$ | 1.5C |
| Geo SSP5-34-OS 2.0 | SSP5-34-OS | 305 Tg $SO_2$ | 2.C |

**Table 2.** Impacts dependent on different meansures: achieved temperature targets applying SAG, amount of sulfur burden, and the baseline scenario.

| Dependencies | Temperature targets (1.5 vs. $2^oC$) | Accumulated $SO_2$ injection | Baseline scenario |
|---|---|---|---|
| Major Importance | Surface air and ocean temperature<br>Land Ice, AMOC<br>Ocean NPP | Global precipitation<br>Ozone hole | Land NPP<br>Ocean acidification |
| Minor Importance | Global precipitation<br>Land NPP<br>Ozone hole | Surface air temperature<br>Land NPP<br>Land ice, AMOC | Surface air ocean temperature<br>Ozone hole<br>Land ice, AMOC |



**Figure 1.** Top panel: Annual surface air temperature evolution for the business as usual case (SSP5-85), the overshoot case that is following business as usual until 2040 and then starting strong mitigation and carbon dioxide removal (SSP5-34-OS), and for 3 different SAG scenarios: based on the SSP5-85 baseline scenario and applying sulfur injections to reduce warming to $1.5^{o}C$ above pre-industrial (PI) conditions (Geo SSP5-85 1.5); based on the SSP5-34-OS and reducing warming to $1.5^{o}C$ above PI (Geo SSP5-34-OS 1.5), and based on the SSP5-34-OS and reducing warming to $2.0^{o}C$ above PI (Geo SSP5-34-OS 2.0) A ten year running mean has been applied to all the timeseries. Black lines indicate the 1850-1900 temperature average (pre-industrial (PI) control temperatures) and the $1.5^{o}C$ and $2.0^{o}C$ surface air temperatures above PI control. Bottom panel: Concentrations of carbon dioxide ($CO_2$), dotted lines, and methan ($CH_4$), solid line, for the 2 baseline simulations.

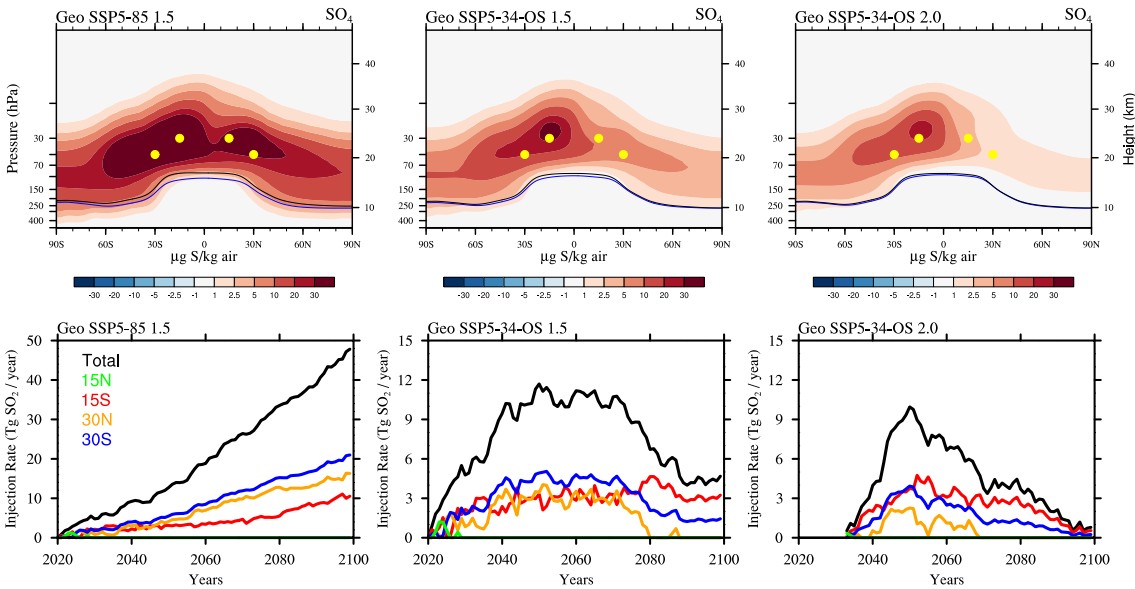

**Figure 2.** Top row: Difference of zonally and annually averaged sulfate $SO_4$ burden between stratospheric sulfur injection cases in 2070–2089 and the control experiment for the same period for Geo SSP5-85 (left), Geo SSP5-34-OS 1.5 (middle), and Geo SSP5-35-OS 2.0 (right). The lapse rate tropopause is indicated as a black line for the control and a blue line for the $SO_2$ injection cases. Yellow dots indicate locations of injection. Bottom row: Injection rate in Tg $SO_2$ per year for the three cases as in the top row: total injections (black), injections at $15^oN$ (green), $15^oS$ (red), $30^oN$ (orange), and $30^oS$ (blue).

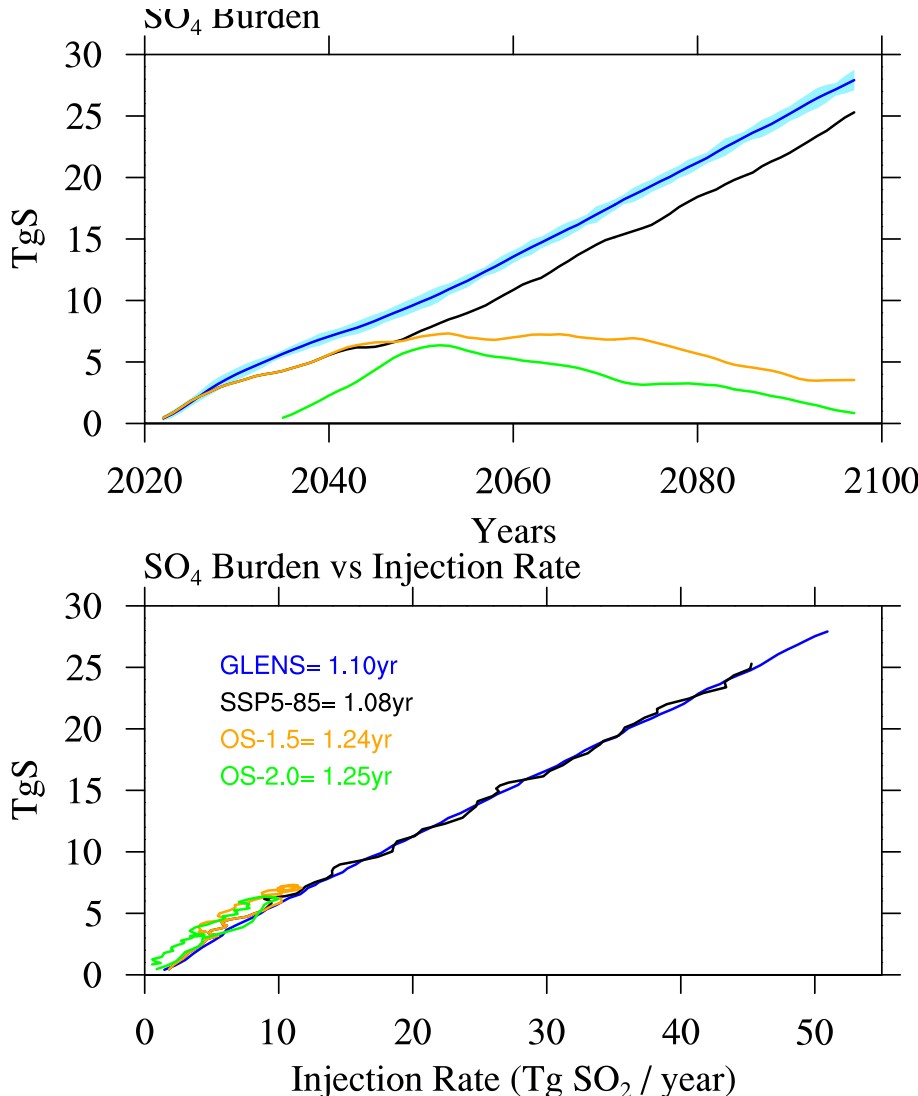

**Figure 3.** Annual averaged stratospheric sulfate aerosol burden in TgS for the geoengineering injection experiments minus the control with time (top panel) and injection rate (bottom panel). The stratospheric sulfate lifetime is listed in the bottom panel. In addition to the model experiments performed in this study, we add result for the Geoengineering Large Ensemble (GLENS). See text more more details.

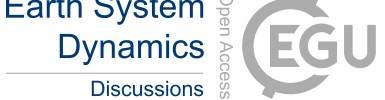



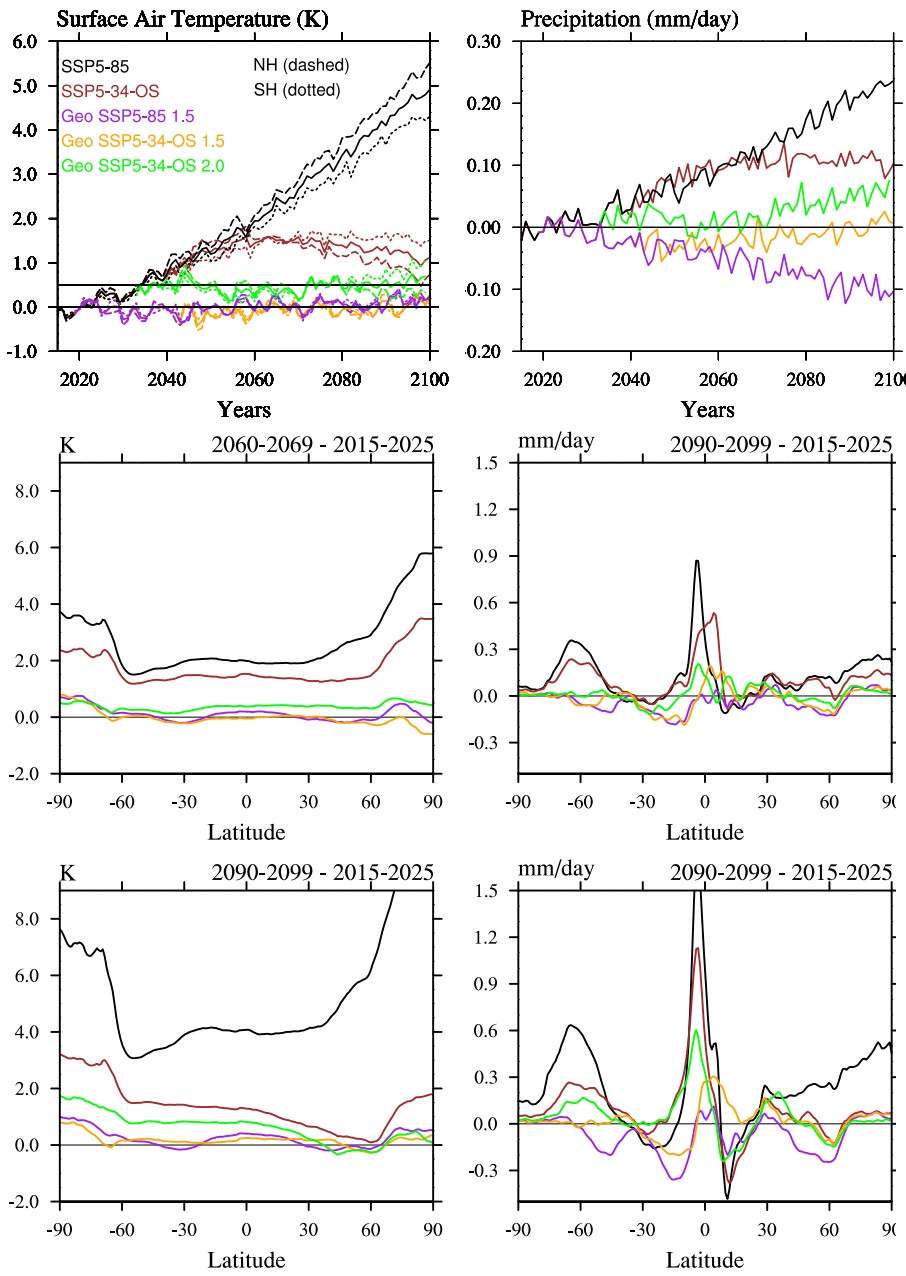

**Figure 4.** Top left: Time evolution of area weighted annual mean surface air temperature with regard to 1850-1900 conditions, averaged over the globe (solid), over the Northern Hemisphere (dashed) and over the Southern Hemisphere (dotted) for different model experiments (different colors, see legend); Top right: Time evolution of area weighted annnual precipitation with regard to 1850-1900 conditions for different model experiments (different colors); Middle and bottom row: differences for zonal mean surface air temperatures (left) and precipitation (right) between values in 2060-2069 (middle) and 2090-99 (bottom) for the different model experiments (different colors) and 2015-2025 SSP5-85 conditions.

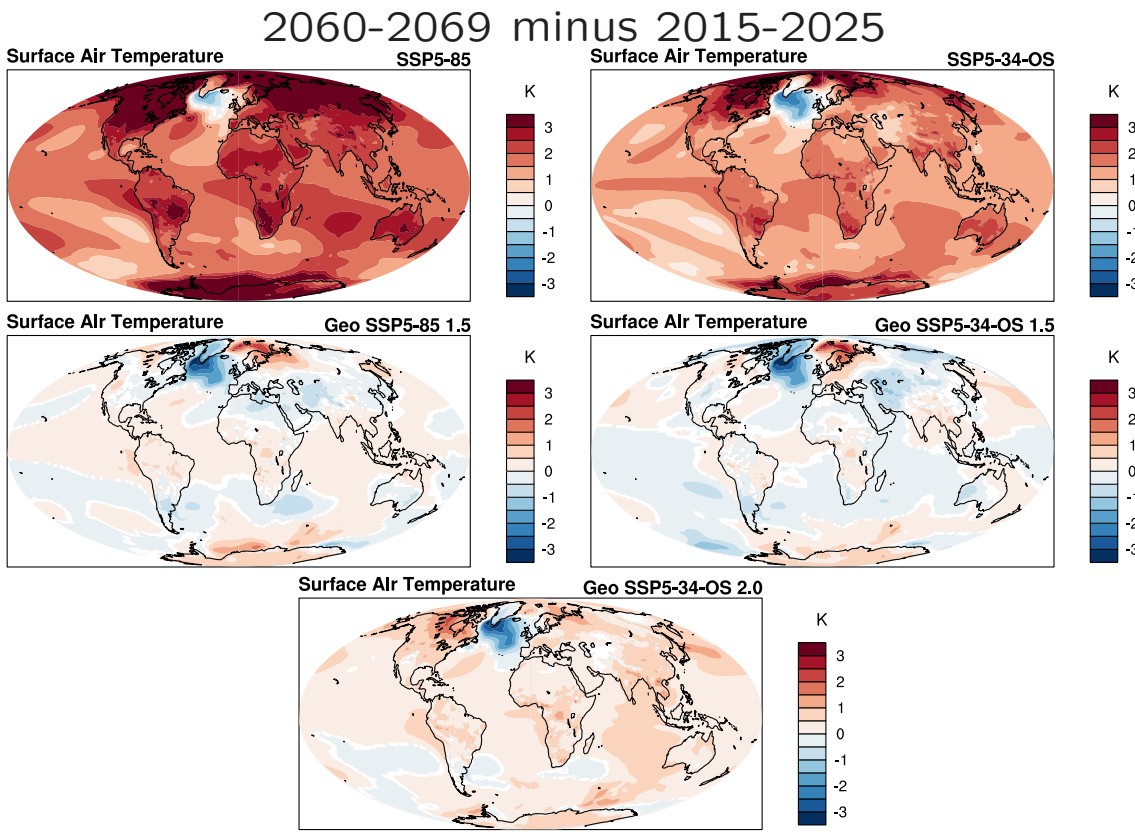

**Figure 5.** Surface Air Temperature difference between 2060–69 and 2015–2025 for SSP5-85 and SSP5-35-OS (top panels), Geo SSP5-85 1.5 and Geo SSP5-34-OS 1.5 (middle panel) and Geo SSP5-35-OS 2.0 (bottom panel).

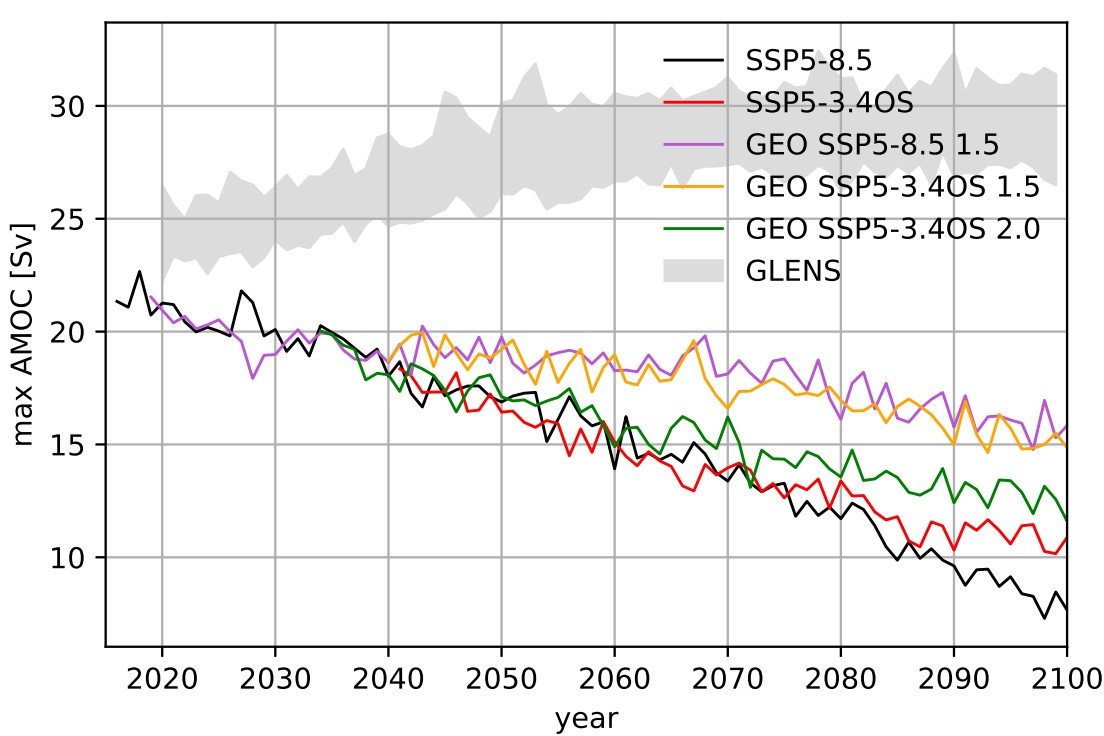

**Figure 6.** Evolution of the maximum North Atlantic Meridional Overturning Circulation strength from the AMOC index for the different scenario's. Shaded grey area is AMOC index range in the 21-member GLENS ensemble. The AMOC index is defined as the maximum flux in the Atlantic Basin between 500m depth to the bottom, and between 28–90°N (Sverdrups).



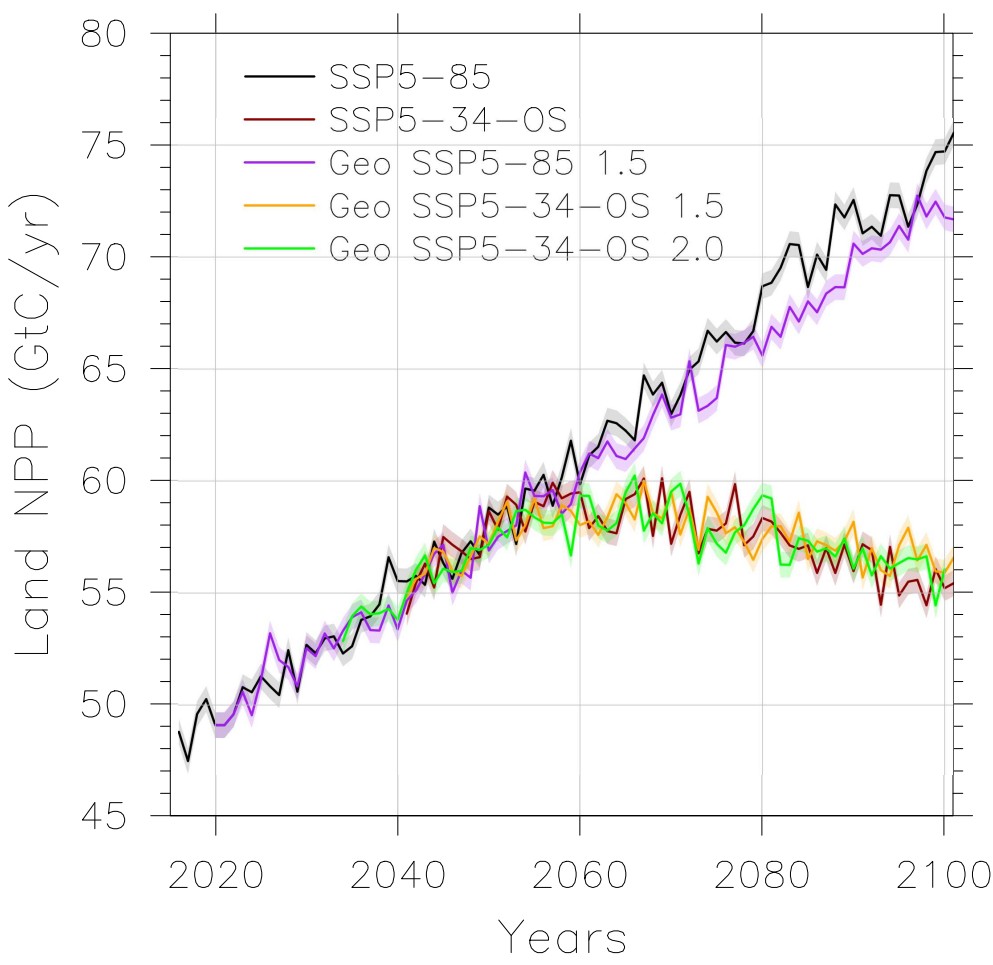

**Figure 7.** Annual land accumulated NPP (GtC/yr) in baseline and SAG scenarios (different colors are indicated in legend). The shaded area is 1 standard deviation of 450 years pre-industrial control simulation.



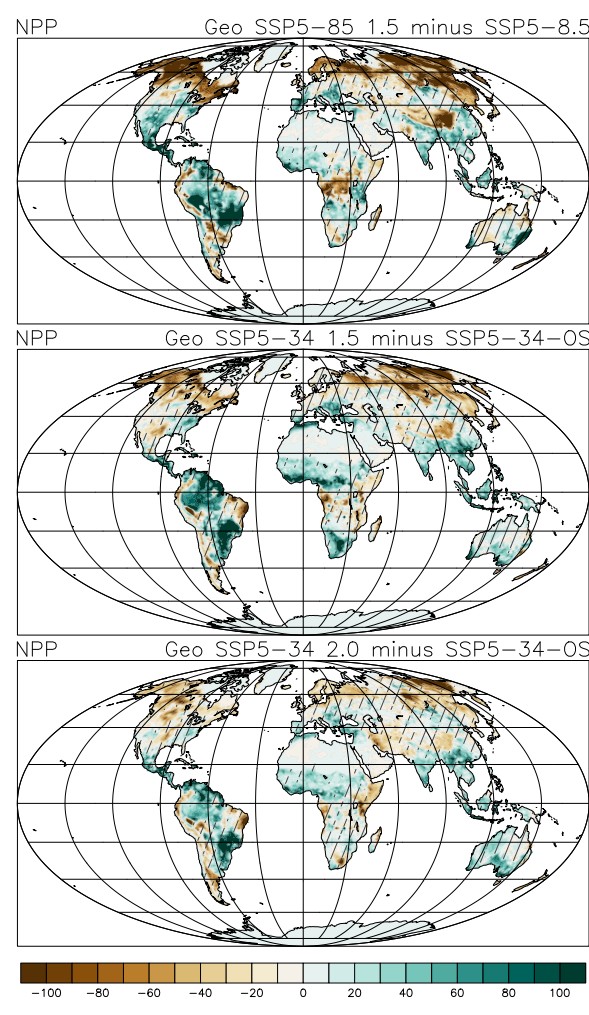

**Figure 8.** Land accumulated NPP difference (gC/m$^2$/yr) between 2060-69 for Geo SSP5-85 1.5 and SSP5-85 (top pannel), Geo SSP5-34-OS 1.5 and SSP5-34-OS (middle pannel), and Geo SSP5-34-OS 2.0 and SSP5-34-OS (bottom pannel). Hatched regions are areas with changes within 1 standard deviation of 450 years pre-industrial control simulation.

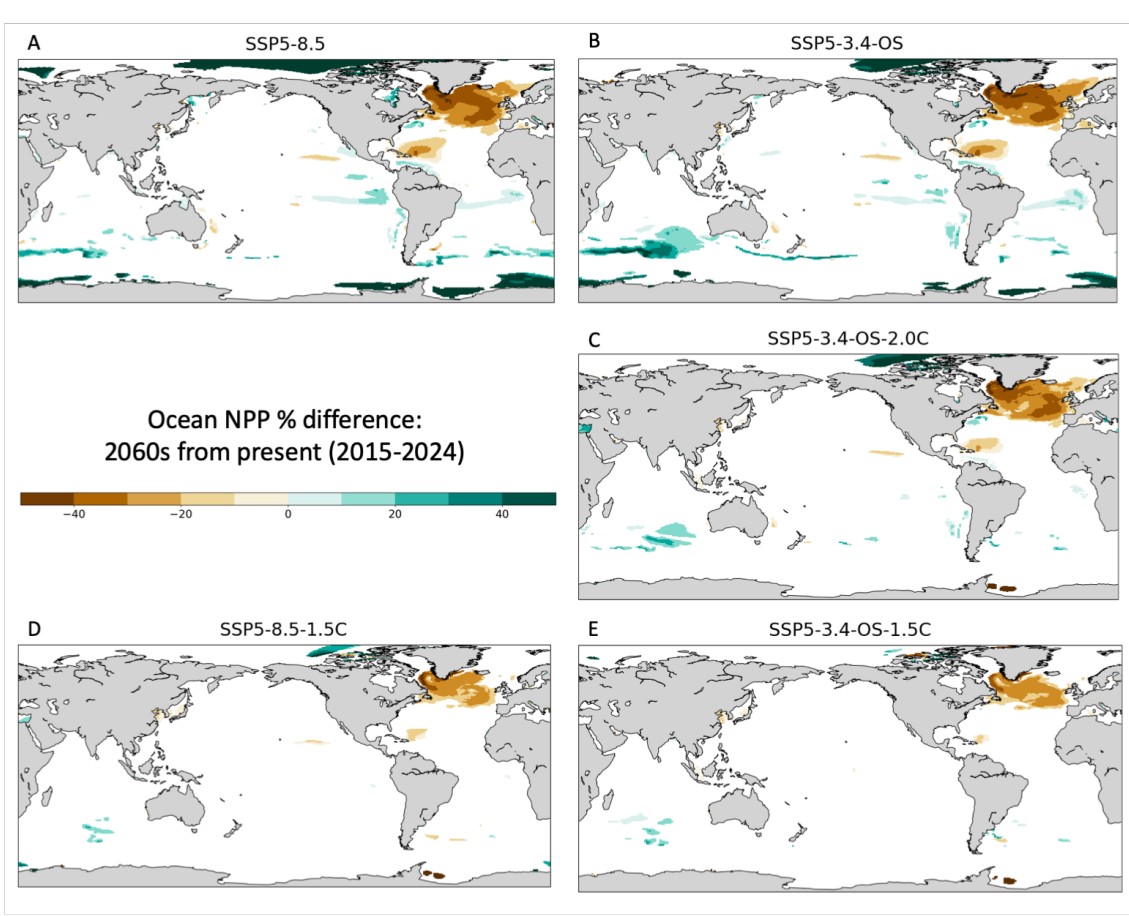

**Figure 9.** Percent difference in ocean net primary productivity (NPP) in 2060-2069 from 2015-2024. Regions shaded in color are significant with 95% confidence.





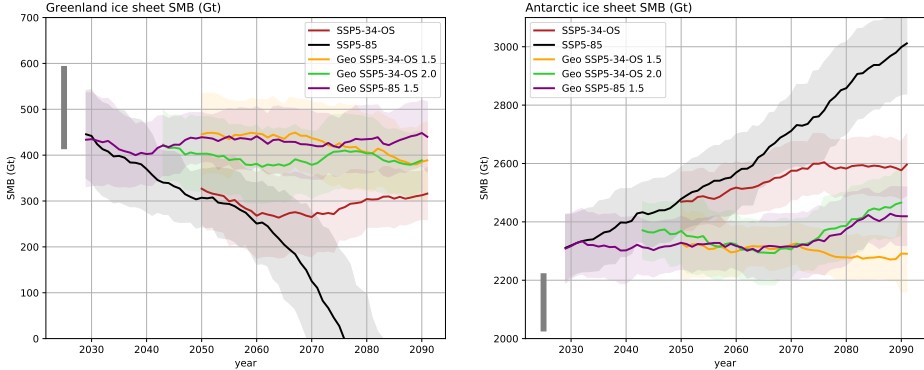

**Figure 10.** Mean ice sheet surface mass balance (SMB) in Gt per year with shading indicating the standard deviation. A 20-year running mean has been applied to filter out year-to-year variability. For the GrIS (left panel), the area of integration is the contiguous ice sheet (1,699,077 km$^2$). For the AIS (right panel), the area of integration is the grounded ice sheet (12,028,595 km$^2$). The solid grey bar indicates the +/- 1 standard deviation SMB over the period 1960-1999 in CESM2(WACCM) for reference.

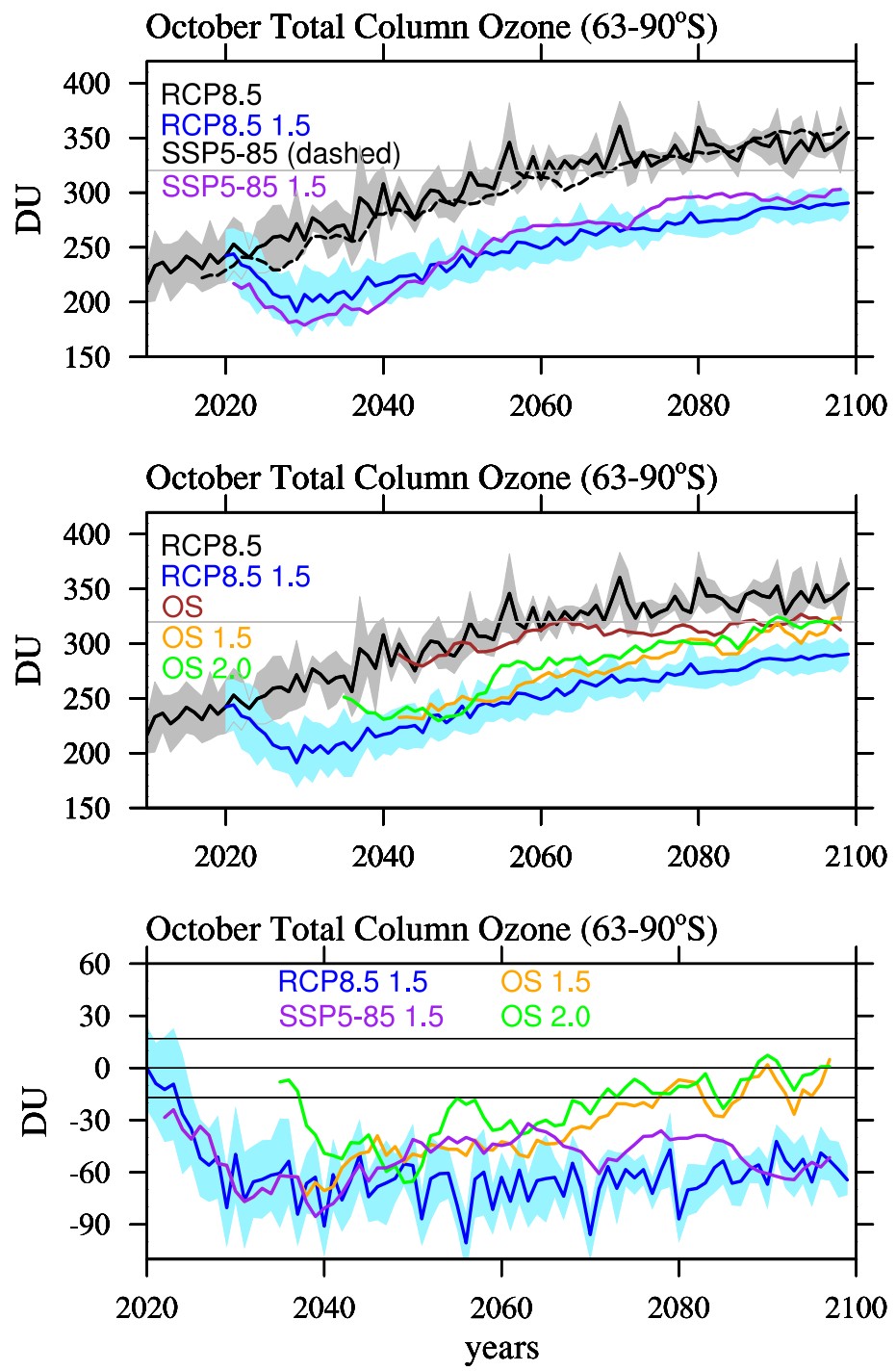

**Figure 11.** Top and middle panel: October averaged total column ozone between 63-90°S for different model experiments (different colors). Grey and and light blue areas show the standard deviation of the GLENS ensemble and the light grey line indicats 1980 values. Bottom panel, differences between geoengineering and control experiments, the two black lines around zero indicate the standard deviation from the GLENS baseline simulations. A running mean over 5 years has been applied to results from the one-member simulations.



**Figure 1.** A1: Simulated sea surface temperature (SST) anomaly and % change in net primary productivity (NPP) in SSP5-8.5 2015–2024 relative to pre-industrial long term means. Regions shaded in color are significant with 95% confidence.





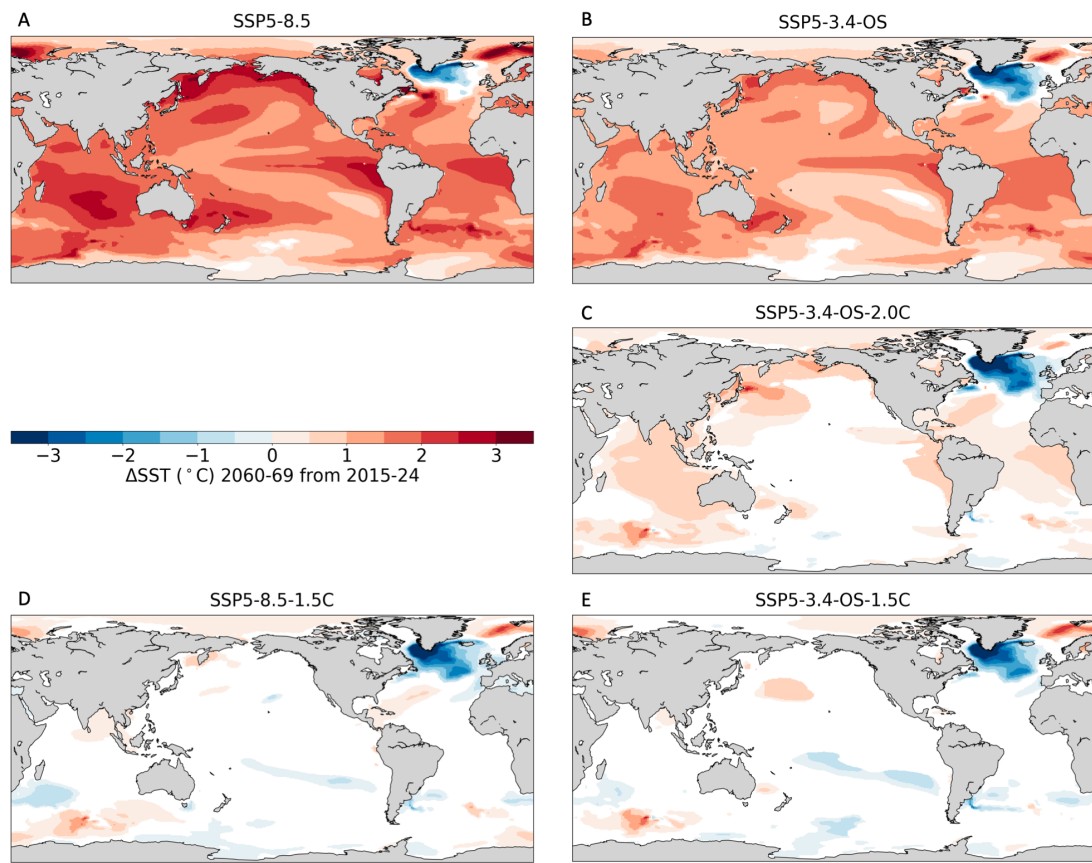

**Figure 2.** A2: Sea surface temperature (SST) in 2060-2069 relative to 2015–2024 for different scenarios (different panels). Regions shaded in color are significant with 95% confidence.





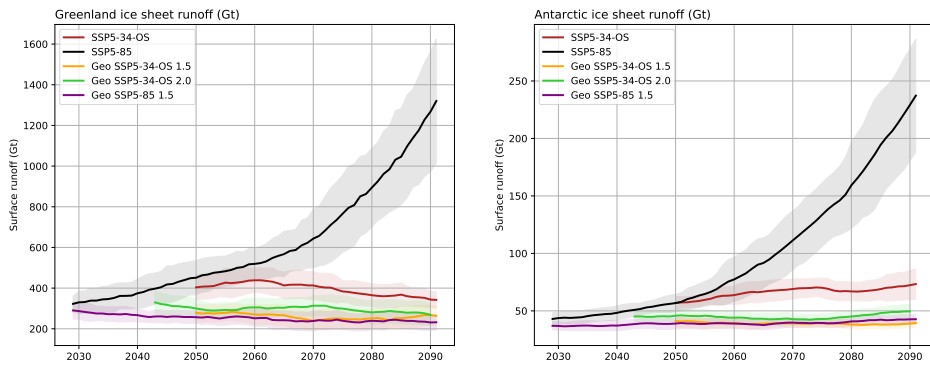

**Figure 3.** A3: Mean ice sheet runoff in Gt per year with shading indicating the standard deviation. A 20-year running mean has been applied to filter out the high year-to-year variability. The area of integration is the same as in Figure 10.