# Peer review of "Reaching 1.5°C and 2.0°C global surface temperature targets using stratospheric aerosol geoengineering"

_Earth System Dynamics, 2019_

## Referee Comment (RC1) · Anonymous Referee #1 · 4 Jan 2020

This study seeks to propose new testbed model experiments for studying scenarios of stratospheric aerosol geoengineering (SAG) designed to limit global warming to fixed global mean surface temperature targets, with some additional constraints to limit undesirable side-effects. I appreciate the interest that the authors' idea has for the earth system community, and I find the paper to be generally well-structured and with a clear logical flow. However, as far as I can tell the main novelties of the study lie in the use of a very recent CMIP6 model, and in the combination of a feedback controller modelling approach with overshoot climate scenarios – neither of which is novel in isolation. Several of the conclusions are likely highly model dependent, and the broader considerations echo the results of other recent studies. Furthermore, some aspects

of the manuscript – for example the figures – have a very unrefined feeling. Finally, I have two major concerns on the structure and contents of the study, which I detail below. Based on this, even though the topic of the study is well-suited to ESD, I am not convinced that it is suitable for publication in this journal.

Major Comments

1. The study performs only one simulation for each geoengineering experiment, citing computational limitations as the main reason. Since the authors state that the study's goal is to establish a protocol for new model experiments, this is justifiable. However, the authors then perform only three SAG experiments; the obvious absent is Geo-SSP85 2.0. Given that comparing SAG interventions with the same temperature goals under different scenarios is a major focus of the study, and that – as the authors themselves underscore – past simulations with earlier model versions show significant differences from the ones presented in this study, I struggle to see the logic in not including such a simulation.

2. The study reads as a generally well-structured, primarily descriptive report of a set of three SAG model simulations. If the aim of the study is indeed to describe new numerical simulations, then I would expect to see a larger number of different experiments, ensembles etc. If, instead, the goal is to establish a protocol for new model experiments, I would expect significant additional analyses and tests on the feedback controller, the latitudes of injection of the aerosols etc. The study is therefore in a grey area between a description of new numerical simulations and a more technical/mechanistic experiment design work, and I find it somewhat unsatisfactory under both categories.

3. I find the figures unsuitable for publication. Some examples: the styles differ across figures and panels within the same figure (e.g. Fig. 4, top row vs. middle and bottom rows), colour-coding/labelling of experiments is inconsistent (e.g. cf. Fig. 1, 3, 11), some figures have panel labels (e.g. Fig. 9), while others do not, different map projections are used (e.g. cf. Figs. 5, 8 and 9) etc. I provide an incomplete list of suggestions
in the minor comments below.

Some Minor Comments

1. Introduction: keeping in mind the relatively broad readership of ESD, it would be useful to add one or two sentences explaining what a "feedback controller" is in this context.

2. Sect. 2.2: echoing the above comment, the description of the feedback control algorithm in this section is poor. Please rephrase and expand it. A practical example of its functioning would be beneficial.

3. p. 2 ll. 44-46 This is a somewhat awkward sentence, please rephrase.

4. p. 6 l. 164 algorithem –> algorithm

5. Sect. 3 The authors use the term "efficiency" in the title, but never refer back to this in the section. I would suggest either discussing this in the text or removing the term altogether.

6. p. 6 l. 184 tropospheric –> troposphere.

7. p. 8 l. 222 "For the baseline simulations, temperatures in high latitudes are higher than in mid and low latitudes" Perhaps the authors mean "temperature anomalies"?

8. p.8 l. 230 "1.5 ËŽC and 2.0 ËŽC" –> 1.5 ËŽC or 2.0 ËŽC

9. Table I: I would suggest adding a column with the models used, as I understand that these vary between the RCP and SSP simulations.

10. All figures: add panel letters to all figures, which makes referencing more straight forward and concise (avoiding sentences like: "Fig. 4, middle and bottom panels on the right").

11. Fig. 1 In the top panel there seems to be a large gap between the SSP scenario and the beginning of the Geo SSP5-34-OS 2.0 experiment. Is that due to the choice of

using RCP8.5 for initialisation? If so, what effects may this have on the results? If not, what is it due to?

12. Fig. 1 Please fix in-panel labels in bottom panel (space between parentheses and "dotted"/"solid").

13. Fig. 2 Top row: the black and blue lines are almost indistinguishable. Please make them thicker, use different line styles, or otherwise modify them to make the figure clearer.

14. Fig. 3 Please move the legend to the top panel.

15. Fig. 3 The title of the top panel is chopped off in the PDF I downloaded.

16. Fig. 3 In the legend, please use full name of the experiments as done in other figures.

17. Fig. 6 Caption: "scenario's" –> "scenarios".

18. Competing interests: "There is are competing interests at present". Barring the "is are", shouldn't these be stated?

---

## Referee Comment (RC2) · Anonymous Referee #2 · 7 Jan 2020

In the present paper the authors propose new testbed experiments for the Geoengineering Model Intercomparison Project. Based on the emission scenarios SSP5-85 and SSP5-34-OS (overshoot scenario) three geoengineering simulation are performed. Stratospheric sulfur injections at different locations are used to limit global warming to 1.5C (for both scenarios) and 2.0C (for SSP5-34-OS). A feedback controller computes the needed amount of sulfur to be injected at different locations to reach temperature targets, and to reduce side effects. The simulations are run utilizing the CESM2(WACCM6) model. Large-scale climate variables are analysed, and some changes are discussed relevant for societal and ecosystem impacts. The results show that some changes depend on the defined temperature targets while others appear to

be sensitive to the amount of the sulfur injections or the CO2 concentration.

Overall I find the paper well-written and well-structured. The designs of the individual scenarios and the analysis of the results are sound. In general, I also appreciate the, according to the authors, main goal of this study: establishing a new model testbed with a clear protocol. In this respect, the proposed scenarios and analyses may serve as a guideline. However, in my view the present version does not provide enough material to warrant publication, and major modifications are needed. Beside a more concrete definition of the proposed testbed (major 1 & 2, minor 2), my main concern is the high level of uncertainty, which may prevent significant interpretation and comparison of the results (major 3).

Major

1) It is not completely clear to me, what the actual proposed testbed (and the related protocol) is. From the abstract it appears (to me) that two simulations limiting global warming to 1.5C or 2.0C (with reducing side effects) based on an overshoot baseline scenario are the central experiments. However, it seems (e.g. discussions and conclusions) that also the comparison with the high greenhouse emission baseline is part of the protocol. This needs clarification. Furthermore, if the latter is true both the 1.5C and the 2.0C case need to be included in this study.

2) The feedback controller appears to me a major factor defining the results, as it determines the sulfur dioxide forcing. It is not clear to me whether the forcing computed by the controller is unique in terms of reaching the given targets and limiting the side effects. A clear defined forcing is, in my view, a major point in defining testbed experiments. In the present case, it seems even more important as some results strongly depend on the particular sulfur injection. Thus, some more words on the forcing (and controller, see minor 3) are needed, in particular: how unique is the forcing obtained from the controller given the set of target temperatures?

3) So far, only one simulation for every scenario has been performed. This strongly

hampers the assessment of uncertainties. For example, it is not clear how much of the pronounced North Atlantic warming hole is related to internal variability or the models sensitivity to the particular forcing. This makes the interpretation of the presented results difficult, and complicates the comparison with simulations performed by other models following the proposed protocol. Thus, without having an (at least very small) ensemble, or any other convincing assessment of the uncertainties, the presented results may not include enough robust information.

Minor & technical

1) It would be valuable to have a more comprehensive motivation for such a testbed. What information may we get from it, except the sensitivity of particular models to a specific forcing scenario which may 'not be policy relevant' (L422)?

2) Independent of my major points above, I think that a protocol as precise as possible would help to establish such a testbed.

3) As the feedback controller appears central for the scenarios and results (see Major 2). Thus, a thorough description would be helpful.

4) Table 1: It may be noted that the RCP-85 simulations are run with a different model version (as far as I understand).

5) L300: citep[]Kravitz2013 -> (Kravitz et al., 2013)

6) L415: SSP5-34_OS 1.5 -> SSP5-34_OS 2.0

7) Figure 3, caption: See text more more -> See text for more

---

## Author Comment (AC1) · 5 Feb 2020

**Response to Anonymous Referee #1**

We thank the Referee 1 for sending very helpful comments and suggestions to the manuscript. All the comments are addressed below in detail:

This study seeks to propose new testbed model experiments for studying scenarios of stratospheric aerosol geoengineering (SAG) designed to limit global warming to fixed global mean surface temperature targets, with some additional constraints to limit un- desirable side-effects. I appreciate the interest that the authors' idea has for the earth system community, and I find the paper to be generally well-structured and with a clear logical flow. However, as far as I can tell the main novelties of the study lie in the use of a very recent CMIP6 model, and in the combination of a feedback controller modelling approach with overshoot climate scenarios – neither of which is novel in isolation. Several of the conclusions are likely highly model dependent, and the broader considerations echo the results of other recent studies. Furthermore, some aspects of the manuscript – for example the figures – have a very unrefined feeling. Finally, I have two major concerns on the structure and contents of the study, which I detail below. Based on this, even though the topic of the study is wellsuited to ESD, I am not convinced that it is suitable for publication in this journal.

Based on the two referees comments we realized that the framing of the paper, on the one hand, proposing a test-bed simulation for GeoMIP and on the other hand, discussing novel findings, did not adequately convey the contributions of the paper – in particular, that the impacts of stratospheric aerosol geoengineering strongly depends on various different aspects of the experiment, the considered baseline scenarios and therefore the CO2 concentrations, the amount of SO2 injection, and the chosen temperature targets. To better emphasize the contributions, in revising the paper we will focus on the novel findings, and use those, along with the potential for model-dependent outcomes, as motivation for the suggestion that these could be new test-bed simulations for GeoMIP. The referee is correct that main novelty of the experiments lies in the fact that we have combined the feedback controller with the overshoot scenario. Other novelties, that we have failed to point out more clearly and plan to mention in the revised manuscript, include that for the first time these types of simulations use the CMIP6 future pathways, which are unique since they are based on socio-economic considerations. Furthermore, we are using an updated Earth-System model, that includes more impact relevant coupling, including interactive crop models, land-ice model, and ocean-bio-chemistry. This paper also serves as an overview paper that describes the general setup of the experiments, while additional papers that are in preparation that will refer to this paper. We also agree with the referee, that the results are highly model dependent. While some of the results are aligned with earlier findings, the focus of the paper is not on repeating what has been done before, but describing potential impact relevant outcomes and for other modeling groups to repeat these experiments in order to produce multi-model comparisons to help determine uncertainties of outcomes. One example is already provided in this paper, comparing the Atlantic Meridional Circulation changes with earlier model results. Further, we have been improving the figures to be added the revised version of the manuscript as suggested by the referee.

The abstract will be modified to support the above points and the main text will be modified accordingly:

New Abstract: "A new set of stratospheric aerosol geoengineering (SAG) model experiments have been performed with CESM2(WACCM6) that are based on the CMIP6 overshoot scenario (SSP5-34-OS) as a baseline scenario to limit global warming to 1.5°C or 2.0°C above 1850-1900 conditions. A feedback algorithm has been used to identify the needed amount of sulfur dioxide injections in the stratosphere at four predefined latitudes, 30°N, 15°N, 15°S, and 30°S, to reach three surface temperature targets: global mean temperature, and inter-hemispheric and poleto-equator temperature gradients. The combination of using an overshoot scenario as a baseline that limits the needed amount of SAG applications and the use of a feedback algorithm to reach pre-defined temperature targets in model experiments is expected to reduce some of the earlier identified side effects of SAG. These experiments are therefore relevant for investigating the impacts on society and ecosystems. Comparisons to SAG simulations based on a high emission pathway baseline scenario (SSP5-85) further help investigate the dependency of impacts using different injection amounts to offset surface warming by SAG. We find that changes from present day conditions (2015-2025) in some variables depend strongly on the defined temperature target (1.5°C vs 2.0°C). These include surface air temperature and related impacts, the Atlantic Meridional Overturning Circulation (AMOC), which impacts ocean net primary productivity, and changes in ice sheet surface mass balance, which impacts sea-level rise. Others, including global precipitation changes and the recovery of the Antarctic ozone hole, depend strongly on the amount of SAG application. Furthermore, land net primary productivity as well as ocean acidification depend mostly on the global atmospheric CO\$ 2\$ concentration and therefore the baseline scenario. Multi-model comparisons of the experiments proposed here would help identify consequences of scenarios that include strong mitigation, carbon dioxide removal with some SAG application, on societal impacts and ecosystems."

**Major Comments**

1. The study performs only one simulation for each geoengineering experiment, citing computational limitations as the main reason. Since the authors state that the study's goal is to establish a protocol for new model experiments, this is justifiable. However, the authors then perform only three SAG experiments; the obvious absent is Geo-SSP85 2.0. Given that comparing SAG interventions with the same temperature goals under different scenarios is a major focus of the study, and that – as the authors themselves underscore – past simulations with earlier model versions show significant differences from the ones presented in this study, I struggle to see the logic in not including such a simulation.

We have now finalized a second ensemble member for each of the simulations presented in the manuscript, and we believe that our findings are more robust with those. Our conclusions in this paper that outcomes of geoengineering strongly depend on different baseline scenarios, injection amount and target temperatures, are supported with the simulations we have presented. Based on the available simulations Geo-SSP5-34-OS 1.5 and Geo-SSP534-OS-2.0, we are able to discuss differences between using different target temperatures. Since computer time is a main issue, we decided to add a second ensemble to the existing experiments, because we think that we can stronger support our conclusions that we could gain from the existing experiments, and not add the additional 2.0 experiment to the high forcing scenario.

For the testbed experiment, we decided to focus on the overshoot scenario. We will explain more clearly that the high forcing scenario is not only performed for comparisons with a high forcing scenario but also to be able to identify difference to the earlier study, using a different model version. In the revised manuscript, we will clarify that the proposed experiments are based on the Geo-SSP5-34-OS baseline scenarios a not on the high forcing scenarios, because this is a more relevant scenario for impact analysis. The overshoot experiment does not require unsustainable amounts of SO2 injections, which provide a potentially more policy-relevant scenario.

2. The study reads as a generally well-structured, primarily descriptive report of a set of three SAG model simulations. If the aim of the study is indeed to describe new numerical simulations, then I would expect to see a larger number of different experiments, ensembles etc. If, instead, the goal is to establish a protocol for new model experiments, I would expect significant additional analyses and tests on the feedback controller, the latitudes of injection of the aerosols etc. The study is therefore in a grey area between a description of new numerical simulations and a more technical/mechanistic experiment design work, and I find it somewhat unsatisfactory under both categories.

We thank the referee for pointing this out. As discussed above, we will shift the focus of the paper to describing and discussing the new numerical simulations and performed a second ensemble member to this study. Based on the findings, we are still planning to recommend that it would be beneficial if the experiments based on the overshoot scenario are performed by other modeling groups to identify the range of outcomes of impact relevant diagnostics. The referee suggests to establish a protocol for new model experiment, additional analysis and tests of the feedback controller are required. We have defined specific of the experiment, including injections at four fixed altitudes at 5 km about the tropopause, and using a feedback controller that will check annual deviations for the defined temperature goals. We do not suggest that modeling groups are using a different setup, since this would complicate analysis in comparing outcomes of different models. However, we agree to provide a better description on details on the implementation of the feedback controller in the revised version of the manuscript as addressed below.

3. I find the figures unsuitable for publication. Some examples: the styles differ across figures and panels within the same figure (e.g. Fig. 4, top row vs. middle and bottom rows), colour-coding/labelling of experiments is inconsistent (e.g. cf. Fig. 1, 3, 11), some figures have panel labels (e.g. Fig. 9), while others do not, different map projections are used (e.g. cf. Figs. 5, 8 and 9) etc. I provide an incomplete list of suggestions in the minor comments below.

We agree with the referee that the figures can be much improved and will apply the same style within figures and use consistent labelling and map projections (including color coding) in the revised version of the manuscript. Changes to the figures pointed out by the referee are shown below. The remaining figures will be completed for the revised version of the manuscript.

**Some Minor Comments**

1. Introduction: keeping in mind the relatively broad readership of ESD, it would be useful to add one or two sentences explaining what a "feedback controller" is in this context.

We agree with the referee and will add more information regarding the feedback controller (both briefly in the introduction, and in more detail in section 2 as suggested below). For the introduction, we will add: "In each year of the simulation, the amount of injection to use at each latitude was adjusted based on the deviations in meeting these goals [the global average surface temperature, as well as the equator-to-pole and interhemispheric temperature gradients, noted in the previous sentence]. In this way, the appropriate injections to use to meet the goals was "learned" as the simulation ran, compensating for uncertainty and avoiding lengthy trial runs.".

2. Sect. 2.2: echoing the above comment, the description of the feedback control algorithm in this section is poor. Please rephrase and expand it. A practical example of its functioning would be beneficial.

We agree with the referee and will add more information on the feedback controller to the revised version of the manuscript so modeling groups can repeat the experiment.

3. p. 2 ll. 44-46 This is a somewhat awkward sentence, please rephrase.

In the revised version of the manuscript we will rephrase the text to:

"GLENS was based on a high forcing future climate scenario (RCP8.5) and required an increasing amount of sulfur injection with time. GLENS simulations have shown that **reaching** global surface temperature and temperature gradient **targets**, results in benefits with respect to temperature related impacts compared to experiments that only **control** for global surface temperature (Kravitz et al., 2019)."

4. p. 6 l. 164 algorithem -> algorithm

We will fix this in the revised version of the manuscript

5. Sect. 3 The authors use the term "efficiency" in the title, but never refer back to this in the section. I would suggest either discussing this in the text or removing the term altogether.

Efficiency in Section 3 has been discussed based on Figure 3, bottom panel. We will clarify this in more detail in the revised version of the manuscript

6. p. 6 l. 184 tropospheric -> troposphere.

Agreed.

7. p. 8 l. 222 "For the baseline simulations, temperatures in high latitudes are higher than in mid and low latitudes" Perhaps the authors mean "temperature anomalies"?

Thanks, we will correct this.

8. p.8 l. 230 "1.5 ËŽC and 2.0 ËŽC" -> 1.5 ËŽC or 2.0 ËŽC

Thanks, we will correct this.

9. Table I: I would suggest adding a column with the models used, as I understand that these vary between the RCP and SSP simulations.

Thanks, we will add an additional column to the Table 1 and add additional numbers for the second ensemble member of each experiment.

10. All figures: add panel letters to all figures, which makes referencing more straight forward and concise (avoiding sentences like: "Fig. 4, middle and bottom panels on the right").

Thanks, we will add letters to the figures.

11. Fig. 1 In the top panel there seems to be a large gap between the SSP scenario and the beginning of the Geo SSP5-34-OS 2.0 experiment. Is that due to the choice of using RCP8.5 for initialisation? If so, what effects may this have on the results? If not, what is it due to?

This is just a plotting error, of not drawing a line between the year of initialization and the first year of output in this simulation.

12. Fig. 1 Please fix in-panel labels in bottom panel (space between parentheses and "dotted"/"solid").

Thanks, we fixed that.

13. Fig. 2 Top row: the black and blue lines are almost indistinguishable. Please make them thicker, use different line styles, or otherwise modify them to make the figure clearer.

The blue and black lines have been made bigger.

14. Fig. 3 Please move the legend to the top panel.

We moved the legend to the top panel, and kept the lifetime information in the bottom panel.

15. Fig. 3 The title of the top panel is chopped off in the PDF I downloaded.

Fixed

16. Fig. 3 In the legend, please use full name of the experiments as done in other figures.

Fixed

17. Fig. 6 Caption: "scenario's" -> "scenarios".

Thanks, we fixed the above.

18. Competing interests: "There is are competing interests at present". Barring the "is are", shouldn't these be stated?

We fix the typo meaning "There are no competing interests at present"

**Updated Figures:**

**Figure 1.** Top panel: Annual surface air temperature evolution for 2 ensemble members of the business as usual case (SSP5-85), the overshoot case that is following business as usual until 2040 and then starting strong mitigation and carbon dioxide removal (SSP5-34-OS), and for 3 different SAG scenarios: based on the SSP5-85 baseline scenario and applying sulfur injections to reduce warming to  $1.5^{\circ}$ C above preindustrial (PI) conditions (Geo SSP5-85 1.5); based on the SSP5-34-OS and reducing warming to  $1.5^{\circ}$ C above PI (Geo SSP5-34-OS 1.5), and based on the SSP5-34-OS and reducing warming to  $2.0^{\circ}$ C above PI (Geo SSP5-34-OS 2.0) A ten year running mean has been applied to all the timeseries. Black lines indicate the 1850-1900 temperature average (pre-industrial (PI) control temperatures) and the  $1.5^{\circ}$ C and  $2.0^{\circ}$ C surface air temperatures above PI control. Bottom panel: Concentrations of carbon dioxide (CO2), dotted lines, and methan (CH4), solid line, for the 2 baseline simulations.

---

## Author Comment (AC2) · 7 Feb 2020

We thank the Referee 2 for sending very helpful comments and suggestions to the manuscript.

The general comments of referee 2 are in line with the comments by referee 1. Therefore, we want to point to the general response we have provided to the first referee: "Based on the two referees comments we realized that the framing of the paper, on the one hand, proposing a test-bed simulation for GeoMIP and on the other hand, discussing novel findings, did not adequately convey the contributions of the paper – in particular, that the impacts of stratospheric aerosol geoengineering strongly depends

on various different aspects of the experiment, the considered baseline scenarios and therefore the CO2 concentrations, the amount of SO2 injection, and the chosen temperature targets. To better emphasize the contributions, in revising the paper we will focus on the novel findings, and use those, along with the potential for model-dependent outcomes, as motivation for the suggestion that these could be new test-bed simulations for GeoMIP. The referee is correct that main novelty of the experiments lies in the fact that we have combined the feedback controller with the overshoot scenario. Other novelties, that we have failed to point out more clearly and plan to mention in the revised manuscript, include that for the first time these types of simulations use the CMIP6 future pathways, which are unique since they are based on socio-economic considerations. Furthermore, we are using an updated Earth-System model, that includes more impact relevant coupling, including interactive crop models, land-ice model, and ocean-bio-chemistry. This paper also serves as an overview paper that describes the general setup of the experiments, while additional papers that are in preparation that will refer to this paper. We also agree with the referee, that the results are highly model dependent. While some of the results are aligned with earlier findings, the focus of the paper is not on repeating what has been done before, but describing potential impact relevant outcomes and for other modeling groups to repeat these experiments in order to produce multi-model comparisons to help determine uncertainties of outcomes. One example is already provided in this paper, comparing the Atlantic Meridional Circulation changes with earlier model results. Further, we have been improving the figures to be added the revised version of the manuscript as suggested by the referee."

Addressing Major Comment 1) It is not completely clear to me, what the actual proposed testbed (and the related protocol) is. From the abstract it appears (to me) that two simulations limiting global warming to 1.5C or 2.0C (with reducing side effects) based on an overshoot baseline scenario are the central experiments. However, it seems (e.g. discussions and conclusions) that also the comparison with the high greenhouse emission baseline is part of the protocol. This needs clarification. Furthermore, if the latter is true both the 1.5C and the 2.0C case need to be included in this

study.

We agree with the referee that the testbed scenarios were not sufficiently explained. We will clarify that the testbed experiment is only based on the overshoot baseline experiment and the additional SSP5-85 cases were performed to be able to compare to earlier studies using a different model version. As described above, we decided to shift the focus of the paper from defining the test bed simulations to describing and discussing the new numerical simulations. We have now finalized a second ensemble member for each of the experiments. We are still planning to recommend that it would be beneficial if the experiments based on the overshoot scenario are performed by other modeling groups to identify the range of outcomes of impact relevant diagnostics.

Addressing Major Comment 2) The feedback controller appears to me a major factor defining the results, as it determines the sulfur dioxide forcing. It is not clear to me whether the forcing computed by the controller is unique in terms of reaching the given targets and limiting the side effects. A clear defined forcing is, in my view, a major point in defining testbed experiments. In the present case, it seems even more important as some results strongly depend on the particular sulfur injection. Thus, some more words on the forcing (and controller, see minor 3) are needed, in particular: how unique is the forcing obtained from the controller given the set of target temperatures?

We agree with the referee to add more information to the controller algorithm and to explain the purpose of the use of this controller: In detail, the controller algorithm is designed to check annual temperatures each year, in order to determine how much SO2 injections are required for each of the four predefined injection locations to reach the 3 temperature targets. Since models will respond differently, it is expected that the amount of SO2 injections will differ for each model version. This has been shown if comparing WACCM6 results with the GLENS results. Therefore, the forcing of SO4 in the stratosphere will differ in each model version, some will require more injections than others, some will require a different amount in different hemisphere. Instead of running a feedback algorithm, the required SO2 injection rates could be estimated through
trial-and-error, but this would be very time-consuming to "learn" the right injection rates to use at multiple latitudes, and as a function of time, to achieve the 3 temperature targets in any given model. We therefore recommend that use of a feedback algorithm, while not an essential component of the testbed specification, is a more efficient way of achieving the desired targets. More details on the feedback controller as described above will be provided in the revised version of the manuscript. We are planning to add an appendix or separate section to the manuscript, including the description on how we implemented the feed forward and how this can be done by other modeling groups. The idea of the proposed GeoMIP testbed experiment is to compare the behavior of different models while the injection rates are chosen to meet the same 3 temperature goals. We are not proposing to compare model results that use the same injection rate but result in the same temperature outcomes. Thus, one question would be to explore how different the forcings will be to reach the same temperature targets. Also, if the same temperature targets have been reached, we can ask the question whether the outcomes on impact relevant measures be different or similar? This approach will help to identify the ranges of outcomes in order to help quantify the ranges of uncertainties. We will also add more details along these line in the revised version of the manuscript.

Addressing Major Comment 3) So far, only one simulation for every scenario has been performed. This strongly hampers the assessment of uncertainties. For example, it is not clear how much of the pronounced North Atlantic warming hole is related to internal variability or the models sensitivity to the particular forcing. This makes the interpretation of the presented results difficult, and complicates the comparison with simulations performed by other models following the proposed protocol. Thus, without having an (at least very small) ensemble, or any other convincing assessment of the uncertainties, the presented results may not include enough robust information.

We have now finalized a second ensemble member for each of the presented experiments and therefore increased the significance of the results. We will show in the revised version of the manuscript that the conclusions drawn from the two ensemble

members have not changed compared to just using the one ensemble member. This is because the variability of the different impact measures between the different ensemble members is to the most part smaller than the difference between the different model experiments.

Addressing Minor & technical: 1) It would be valuable to have a more comprehensive motivation for such a testbed. What information may we get from it, except the sensitivity of particular models to a specific forcing scenario which may 'not be policy relevant' (L422)?

We agree with the referee and add more information to the motivation of these experiments, as discussed in the response to Major comment 2.

2) Independent of my major points above, I think that a protocol as precise as possible would help to establish such a testbed.

We agree with the referee and will describe a precise protocol to allow other modeling groups to perform the same experiments. We will focus only on the cases using the overshoot scenario and clarify this.

3) As the feedback controller appears central for the scenarios and results (see Major 2). Thus, a thorough description would be helpful.

In the revised version of the manuscript, we are planning to provide detailed information to allow modeling groups to implement the feedback algorithm.

4) Table 1: It may be noted that the RCP-85 simulations are run with a different model version (as far as I understand).

As suggested by referee 1, we will add an additional column to clarify the model versions used for the different experiments.

5) L300: citep[]Kravitz2013 -> (Kravitz et al., 2013) 6) L415: SSP5-34_OS 1.5 -> SSP5-34_OS 2.0 7) Figure 3, caption: See text more more -> See text for more

We will correct the 3 items above.

---

## Author Response (AR1)

Response to Anonymous Referee #1

We thank the Referee 1 for sending very helpful comments and suggestions to the manuscript. All the comments are addressed below in detail:

*This study seeks to propose new testbed model experiments for studying scenarios of stratospheric aerosol geoengineering (SAG) designed to limit global warming to fixed global mean surface temperature targets, with some additional constraints to limit undesirable side-effects. I appreciate the interest that the authors' idea has for the earth system community, and I find the paper to be generally well-structured and with a clear logical flow. However, as far as I can tell the main novelties of the study lie in the use of a very recent CMIP6 model, and in the combination of a feedback controller modelling approach with overshoot climate scenarios – neither of which is novel in iso- lation. Several of the conclusions are likely highly model dependent, and the broader considerations echo the results of other recent studies. Furthermore, some aspects of the manuscript – for example the figures – have a very unrefined feeling. Finally, I have two major concerns on the structure and contents of the study, which I detail below. Based on this, even though the topic of the study is well-suited to ESD, I am not convinced that it is suitable for publication in this journal.*

Based on the two referees comments we realized that novel findings of this paper have not been adequately conveyed. We have significantly improved the revised version of the manuscript to address those concerns. The referee is correct that main novelty of the experiments lies in the fact that we have combined the feedback controller with the overshoot scenario. In the revised version of the paper we improve the motivation for these experiments, which are more important for assessing social and ecological relevant impacts than previous GeoMIP experiments. Furthermore, findings of the paper are pointed out more clearly, e.g., impacts of stratospheric aerosol geoengineering strongly depend on various different aspects of the experimental design, including the considered baseline scenarios and therefore the $CO_2$ concentrations, the amount of $SO_2$ injection, and the chosen temperature targets. Other novelties, include that the experiments are based on the CMIP6 future pathways, which are unique since they are based on socio-economic considerations. We agree with the referee that the results are highly model dependent and it is therefore important to produce multi-model comparisons to help determine uncertainties of impact relevant measures. The paper has been revised and in particular the abstract and discussion and conclusions have been modified accordingly. We improved the figures in the revised version of the manuscript as suggested by the referee.

*Major Comments*

*1. The study performs only one simulation for each geoengineering experiment, citing computational limitations as the main reason. Since the authors state that the study's goal is to establish a protocol for new model experiments, this is justifiable. However, the authors then perform only three SAG experiments; the obvious absent is Geo-SSP85 2.0. Given that comparing SAG interventions with the same temperature goals under different scenarios is a major focus of the study, and that – as the authors themselves underscore – past simulations with earlier model versions show significant differences from the ones presented in this study, I struggle to see the logic in not including such a simulation.*

In the revised version of the paper we have clarified that we propose SRM experiments that are based on the overshoot scenario, comparing the two different temperature targets. The additional comparisons to the high forcing scenario have been performed to help to identify impacts using a different baseline scenario and to identify differences with the earlier study, using a different model

version. We don't think that the addition of the Geo-SSP85 2.0 scenario would add much to the conclusions. Based on the available simulations Geo-SSP5-34-OS 1.5 and Geo-SSP534-OS-2.0, we are able to discuss differences between different target temperatures using the same baseline scenario. We have now finalized a second ensemble member for each of the simulations presented in the manuscript, and we believe that our findings are more robust with those.

*2. The study reads as a generally well-structured, primarily descriptive report of a set of three SAG model simulations. If the aim of the study is indeed to describe new numerical simulations, then I would expect to see a larger number of different experiments, ensembles etc. If, instead, the goal is to establish a protocol for new model experiments, I would expect significant additional analyses and tests on the feedback controller, the latitudes of injection of the aerosols etc. The study is therefore in a grey area between a description of new numerical simulations and a more technical/mechanistic experiment design work, and I find it somewhat unsatisfactory under both categories.*

We thank the referee for pointing this out. We have shifted the focus of the paper to describing and discussing the new numerical simulations and performed a second ensemble member to this study. Based on the findings, we are still recommending that it would be beneficial if the experiments based on the overshoot scenario are performed by other modeling groups to identify the range of outcomes of impact relevant diagnostics.

The referee suggests that to establish a protocol for new model experiment, additional analysis and tests of the feedback controller are required. We agree with the reviewer that more tests of the feedback controller in terms of latitudes and altitudes would be required to be refined. However, as has been shown in the earlier studies, the defined setup has been successful to reach the pre-defined temperature targets. For comparisons of different model results, it is important to use the same protocol. In the revised version of the manuscript, we have more clearly defined the specifics of the experiment, including requiring injections at four fixed altitudes at 5 km about the tropopause, and using a feedback controller that will check annual deviations for the defined temperature goals. We provided more details on the feedback control algorithm in the appendix and added a figure. We suggest that modeling groups use the same setup and defined surface temperature targets, to be able to directly compare the outcomes of different models.

*3. I find the figures unsuitable for publication. Some examples: the styles differ across figures and panels within the same figure (e.g. Fig. 4, top row vs. middle and bottom rows), colour-coding/labelling of experiments is inconsistent (e.g. cf. Fig. 1, 3, 11), some figures have panel labels (e.g. Fig. 9), while others do not, different map projections are used (e.g. cf. Figs. 5, 8 and 9) etc. I provide an incomplete list of suggestions in the minor comments below.*

The listed figures have been improved and apply now the same style within figures and use consistent labelling and map projections (including color coding) in the revised version of the manuscript.

*Some Minor Comments*

*1. Introduction: keeping in mind the relatively broad readership of ESD, it would be useful to add one or two sentences explaining what a "feedback controller" is in this context.*

We agree with the referee and added to the introduction: "The experiments used a feedback controller to maintain global average surface temperatures, as well as equator-to-pole and interhemispheric temperature gradients, at 2020 levels. After each year of the simulation, the amount of sulfur injections at each of the four different latitude locations in the stratosphere was calculated, based on the deviations in meeting these surface temperature goals (see Appendix for more details).

*2. Sect. 2.2: echoing the above comment, the description of the feedback control algorithm in this section is poor. Please rephrase and expand it. A practical example of its functioning would be beneficial.*

We agree and added a new section in the appendix on the feedback control algorithm.

*3. p. 2 ll. 44-46 This is a somewhat awkward sentence, please rephrase.*

In the revised version of the manuscript we will rephrase the text to:

"GLENS was based on a high forcing future climate scenario (RCP8.5) and required an increasing amount of sulfur injection with time. GLENS simulations have shown that using global surface temperature and surface temperature gradients as targets, instead of only controlling for global surface temperature, results in reduced side effects, including more even cooling and reduced shifts in precipitation pattern (Kravitz et al., 2019)."

*4. p. 6 l. 164 algorithem –> algorithm*

changed

*5. Sect. 3 The authors use the term "efficiency" in the title, but never refer back to this in the section. I would suggest either discussing this in the text or removing the term altogether.*

Thanks for this comment. We remove this term from the title and clarify that we are discussing aerosol burden with regard to sulfur injections per years, based on Figure 3. To clarify, the following sentences have been revised:

"Differences between the 3 SAG experiments and the Geo RCP85 1.5 also arise in terms of accumulated $SO_2$ injection amount (Table 1)  and aerosol burden with regard to sulfur injections per year (Fig. 3).

And later:

"Both experiments that are based on the OS baseline scenario show larger burden per injection amount (Fig. 3, panel b) for the years when $SO_2$ injections have been declining because of the prevalent sulfate burden from previous years."

*6. p. 6 l. 184 tropospheric –> troposphere.*

changed

*7. p. 8 l. 222 "For the baseline simulations, temperatures in high latitudes are higher than in mid and low latitudes" Perhaps the authors mean "temperature anomalies"?*

Yes, we have corrected this.

*8. p.8 l. 230 "1.5C and 2.0C" –> 1.5C or 2.0C*

This has been corrected

*9. Table I: I would suggest adding a column with the models used, as I understand that these vary between the RCP and SSP simulations.*

Thanks, we added additional column to the Table 1 and additional numbers for the second ensemble member of each experiment.

*10. All figures: add panel letters to all figures, which makes referencing more straight forward and concise (avoiding sentences like: "Fig. 4, middle and bottom panels on the right").*

Thanks, we added letters to all multiple panel figures.

*11. Fig. 1 In the top panel there seems to be a large gap between the SSP scenario and the beginning of the Geo SSP5-34-OS 2.0 experiment. Is that due to the choice of using RCP8.5 for initialisation? If so, what effects may this have on the results? If not, what is it due to?*

This was a plotting error of not drawing a line between the year of initialization and the first year of output in this simulation, which has been fixed.

*12. Fig. 1 Please fix in-panel labels in bottom panel (space between parentheses and "dotted"/"solid").*

Thanks, we fixed that.

*13. Fig. 2 Top row: the black and blue lines are almost indistinguishable. Please make them thicker, use different line styles, or otherwise modify them to make the figure clearer.*

The blue and black lines have been made thicker.

*14. Fig. 3 Please move the legend to the top panel.*

We moved the legend to the top panel, and kept the lifetime information in the bottom panel.

*15. Fig. 3 The title of the top panel is chopped off in the PDF I downloaded.*

This has been fixed

*16. Fig. 3 In the legend, please use full name of the experiments as done in other figures.*

This has been fixed

*17. Fig. 6 Caption: "scenario's" –> "scenarios".*

Thanks, this has been fixed.

*18. Competing interests: "There is are competing interests at present". Barring the "is are", shouldn't these be stated?*

We fix the typo meaning "There are no competing interests at present"

Response to Anonymous Referee #2

*Overall I find the paper well-written and well-structured. The designs of the individual scenarios and the analysis of the results are sound. In general, I also appreciate the, according to the authors, main goal of this study: establishing a new model testbed with a clear protocol. In this respect, the proposed scenarios and analyses may serve as a guideline. However, in my view the present version does not provide enough material to warrant publication, and major modifications are needed. Beside a more concrete definition of the proposed testbed (major 1 & 2, minor 2), my main concern is the high level of uncertainty, which may prevent significant interpretation and comparison of the results (major 3).*

We thank Referee 2 for sending very helpful comments and suggestions to the manuscript. Based on the two referee's comments we realized that novel findings of this paper have not been adequately conveyed and have significantly improved the revised version of the manuscript accordingly, as outlined below in detail.

*Major comments*

*1) It is not completely clear to me, what the actual proposed testbed (and the related protocol) is. From the abstract it appears (to me) that two simulations limiting global warming to 1.5C or 2.0C (with reducing side effects) based on an overshoot baseline scenario are the central experiments. However, it seems (e.g. discussions and conclusions) that also the comparison with the high greenhouse emission baseline is part of the protocol. This needs clarification. Furthermore, if the latter is true both the 1.5C and the 2.0C case need to be included in this study.*

We agree with the referee that the text, in particular the discussions and conclusions, were misleading and we have revised these parts. We clarified in the new version of the manuscript that the testbed experiment is indeed based only on the overshoot baseline experiment. The additional SSP5-85 cases were performed to be able to compare to earlier studies using a different model version and to analyze the effects if using a different baseline simulation. We revised the description of the experiment in Section 2.2.

*Addressing Major Comment 2) The feedback controller appears to me a major factor defining the results, as it determines the sulfur dioxide forcing. It is not clear to me whether the forcing computed by the controller is unique in terms of reaching the given targets and limiting the side effects. A clear defined forcing is, in my view, a major point in defining testbed experiments. In the present case, it seems even more important as some results strongly depend on the particular sulfur injection. Thus, some more words on the forcing (and controller, see minor 3) are needed, in particular: how unique is the forcing obtained from the controller given the set of target temperatures?*

We agree with the referee and added more information to the controller algorithm in the introduction and added text and a figure in the appendix and point to the important references: In detail, the controller algorithm is designed to check annual temperatures each year, in order to determine how much $SO_2$ injection is required for each of the four predefined injection locations to reach the 3 temperature targets. Since models will respond differently, it is expected that the amount of $SO_2$ injection will differ for each model version. This has been shown if comparing WACCM6 results with the GLENS results. Therefore, the forcing of $SO_4$ in the stratosphere will differ in each model version, some will require more injections than others, some will require a different amount in different hemispheres.

The idea of the proposed GeoMIP testbed experiment is to compare the behavior of different models while the injection rates are chosen to meet the same 3 temperature goals. We are not proposing to compare model results that use the same injection rate but ones that result in the same temperature outcomes.  Thus, one question would be to explore how different the forcings will be to reach the same temperature targets. Also, if the same temperature targets have been reached, we can ask the question whether the outcomes on impact relevant measures be different or similar? This approach will help to identify the ranges of outcomes in order to help quantify the ranges of uncertainties.

Instead of running a feedback algorithm, the required $SO_2$ injection rates could be estimated through trial-and-error, but this would be very time-consuming to "learn" the right injection rates to use at multiple latitudes, and as a function of time, to achieve the 3 temperature targets in any given model. We therefore recommend that use of a feedback algorithm, while not an essential component of the testbed specification, is a more efficient way of achieving the desired targets.

*Addressing Major Comment 3) So far, only one simulation for every scenario has been performed. This strongly hampers the assessment of uncertainties. For example, it is not clear how much of the pronounced North Atlantic warming hole is related to internal variability or the models sensitivity to the particular forcing. This makes the interpretation of the presented results difficult, and complicates the comparison with simulations performed by other models following the proposed protocol. Thus, without having an (at least very small) ensemble, or any other convincing assessment of the uncertainties, the presented results may not include enough robust information.*

We have now finalized a second ensemble member for each of the presented experiments and therefore increased the significance of the results. Conclusions drawn from the two ensemble members have not changed significantly compared to just using the one ensemble member. This is because the variability of the different impact measures between the different ensemble members is to the most part smaller than the difference between the different model experiments.

*Addressing Minor & technical:*

*1) It would be valuable to have a more comprehensive motivation for such a testbed. What information may we get from it, except the sensitivity of particular models to a specific forcing scenario which may 'not be policy relevant' (L422)?*

We agree with the referee and we added more information to the motivation of these experiments at different places in the paper, including in the abstract:

"The combination of using an overshoot scenario as a baseline that limits the needed amount of SAG applications and the use of a feedback algorithm to reach pre-defined temperature targets in model experiments is expected to reduce some of the earlier identified side effects of SAG. These experiments are therefore relevant for investigating the impacts on society and ecosystems."

In the introduction:

We further establish a protocol for new GeoMIP testbed experiments that are designed to reach 1.5°C and 2.0°C surface temperature targets and are based on the SSP5-34-OS scenario in order to require less sulfur injection than using a high forcing scenario. We require the use of four pre-defined stratospheric

injection locations as well as the use of a feedback controller (or a similar approach) to keep global surface temperatures, inter-hemispheric and pole-to-equator surface temperatures, at the defined target temperatures. These experiments are more relevant for impact analysis than any of the existing GeoMIP experiments. We hope to motivate other modeling groups to conduct the same experiments, thereby allowing for an analysis of the outcomes from a multi-model perspective.

And in the discussions and conclusions:

"Both limited applications of SAG and improved climate targets result in reduced climate impacts and risks, and are therefore better suited for studying impacts on society and ecosystems than much larger scale SAG applications. Multi-model experiments are needed to identify the range of outcomes and uncertainties. We therefore recommend including these experiments as a new testbed GeoMIP scenario for CMIP6. "

*2) Independent of my major points above, I think that a protocol as precise as possible would help to establish such a testbed.*

We agree with the referee and revised Section2.2 to clarify the protocol.

*3) As the feedback controller appears central for the scenarios and results (see Major 2). Thus, a thorough description would be helpful.*

In the revised version of the manuscript, we added more detailed information in the appendix.

*4) Table 1: It may be noted that the RCP-85 simulations are run with a different model version (as far as I understand).*

As suggested by referee 1, we added an additional column to clarify the model versions used for the different experiments.

*5) L300: citep[]Kravitz2013 -> (Kravitz et al., 2013)*

Has been corrected.
*6) L415: SSP5-34_OS 1.5 -> SSP5-34_OS 2.0*

Has been corrected.

*7) Figure 3, caption: See text more more -> See text for more*

Has been corrected.

[revised manuscript text omitted]

---

## Author Response (AR2)

**Review Response**
We thank both reviewers for reviewing the paper a second time and sending additional comments. Please see our responses below.

**Reviewer 1**

*The authors have provided a reasoned reply to my original set of comments, and I find that the overall message is now much clearer than in the previous version of the paper. I also appreciated the addition of some technical details, such as in Appendix A1. However, there are still a number of inaccuracies and minor issues that I would suggest the authors to address before the paper may be published in ESD.*

*1. In the abstract, the authors state that: ”The combination of using an overshoot scenario as a baseline that limits the needed amount of SAG applications and the use of a feedback algorithm to reach predefined temperature targets in model experiments is expected to reduce some of the earlier identified side effects of SAG.” This is a somewhat ambiguous statement, as it places on the same level the feedback algorithm and the choice of scenario. However, the feedback algorithm is an implementation method, while the choice of scenario is a somewhat arbitrary assumption on our future social, political and economic development. I think it is important to clearly distinguish between the two, and concisely motivate in the abstract the relevance of using an overshoot scenario. To me, this relevance is not that it limits the needed amount of SAG applications – in that case the most logical scenario to use would be SSP1-1.9 – but rather that it is conducive to investigating a peak-shaving scenario, which is a potentially effective proposal for how to deal with the ongoing rapid climate change.*

We thank the reviewer for the comment. The reviewer is correct that there are two separate issues addressed in this paper that both are expected to reduce side effects. One is the use of the overshoot as the baseline scenario, while the other is reaching the three temperature targets (though the feedback controller) to reduce additional side effects. The idea here is really to apply those two together. While it is relevant to limit the duration of the SAG application through a peak-shaving application, as the reviewer is pointing out, limiting the amount of SAG is also important to reduce side effects, as shown in this paper.

To address the comment and to more clearly distinguish between the two we changed the first part of the abstract to:
“A new set of stratospheric aerosol geoengineering (SAG) model experiments have been performed with CESM2(WACCM6) that are based on the CMIP6 overshoot scenario (SSP5-34-OS) as a baseline scenario to limit global warming to 1.5°C or 2.0°C above 1850–1900 conditions. The overshoot scenario allows us to apply a peak-shaving scenario that reduces the needed duration and amount of SAG application compared to a high forcing scenario. In addition, a feedback algorithm identifies the needed amount of sulfur dioxide injections in the stratosphere at four predefined latitudes, 30°N, 15°N, 15°S, and 30°S, to reach three surface temperature targets: global mean temperature, and interhemispheric and pole-to-equator temperature gradients. These targets further help to reduce side effects, including overcooling in the tropics, warming of high latitudes and large shifts in precipitation patterns.”

*2. In the abstract, the authors write:” Others, including global precipitation changes and the*

*recovery of the Antarctic ozone hole, depend strongly, on the amount of SAG application". But is it not the case that they depend: "on the amount and details of SAG application"?*

We agree that the effects of SAG on precipitation depend on various other factors. The statement in the paper is not excluding the fact that other factors also play a role.

*3. Related to (1) the above, in the introduction I would discuss the relation between the overshoot scenario and peak shaving in the paragraph starting on l. 55, instead of the one starting on l. 68. Else the reader in the latter paragraph doesn't immediately understand the logic of using an OS scenario and how this combines "two main objectives that have only been addressed separately in previous studies".*

We agree with the suggestion by the reviewer and added information to the overshoot scenario earlier in the text:

"Several studies have pointed out that SAG may be able to reduce some of the effects of global warming temporarily while decarbonization efforts (including mitigation and negative emissions through carbon dioxide removal) are ramped up. **This would be possible if following an overshoot scenario, where mitigation and decarbonization is applied in a way that surface temperatures would peak above desired temperature targets for a limited time and then slowly decline below these temperature targets.** A so-called peak-shaving scenario using SAG was proposed that would potentially help prevent reaching tipping points until greenhouse gas levels have been sufficiently reduced."

*4. l. 137 I don't quite understand this sentence: "While different ensemble members may reach the target temperature at different times, they still have to be setup to reach the same temperature targets and may start at the same time." What do the authors mean by "start at the same time"? Do they refer to the SAG intervention? Please rephrase.*

To clarify this part, we rephrased the text:

"The start of each climate intervention experiment is defined by the time that the baseline simulation has reached a near-surface global-mean temperature of 1.5°C and 2.0°C above pre-industrial, considering a ten-year running mean (in WACCM6 this is around 2020–2025 for 1.5°C and around 2034 for 2°C). **For easier comparisons, all ensemble members use the same exact numbers for the three temperature targets, even if surface temperatures slightly vary for different ensemble members.**"

*5. l. 144 Is there a specific reason for picking 180° W or is longitude largely irrelevant and just an arbitrary choice? Either way, please specify this in the text.*

There is no specific reason for picking 180° W. It just has to be at one point and this was also chosen by the GLENS simulations. We changed the text to:

"Sulfur dioxide injections into the stratosphere are performed at 4 locations 5 km above the tropopause, at 15°N, 15°S, 30°N, and 30°S in latitude, and **at an arbitrary longitude of 180°W**, following the approach described in Kravitz et al. (2017) and Tilmes et al. (2018)."

*6. l. 150 The authors mention the SSP-8.5 scenario here, but only describe the relevant experiment in the paragraph starting on l. 161. It may be more logical to invert the order of the two and first describe all experiments conducted here and then describe the feedback controller and temperature gradient constraints.*

Based on the first round of reviews, we have moved the description of the SSP5-85 scenario to the end of this section, to distinguish from the simulations that are part of the proposed "testbed" experiments. We therefore prefer to keep it the way it is now.

*7. l. 333 Note that this only holds in the case of a perfectly Gaussian distribution. A more robust approach would be using percentiles of the single-gridpoint distributions.*

The reviewer is absolutely correct that the 95% confidence interval language is only robust for normally distributed data. The text has been modified to make this clear.
"Anomalies outside historical climate variability are one indication of ocean conditions that ecosystems are not adapted to, and thus expected to cause disruption to fisheries and natural ecosystems (Bopp et al., 2013; Heneghan et al., 2019). Accordingly, the significance of SST (Fig. A3) and NPP (Fig. 9) anomalies was determined by using the standard deviation ($\sigma$) in each model grid cell of the yearly means from the 499-year pre-industrial control run. An anomaly was considered significant when it was greater than 1.96 $\sigma$ (95% confidence interval) **assuming normally distributed data**."

We are already using a threshold for significance higher or similar to many other studies (e.g., Fasulo et al. 2018; Brady et al., 2017, Krumhardt et al., 2017). While we agree that a percentile analysis would be interesting, it would not change the results very much here, and definitely would not change the messaging of the paper. If one was going to engage in a detailed statistical analysis relevant for ecosystems, it would make more sense to go into monthly or even daily data and look for extreme climate events and phenology changes, which is well beyond the scope here. We agree ours is a somewhat crude statistical analysis, especially from the point of view of ecological studies, but that is what we were going for in this paper, using a common statistical method in the climate modeling community. We look forward to engaging in more nuanced analysis in future in more ecologically-focused studies.

*8. Sect. 4.6/Fig. 10b The offset in the starting point of the Antarctic SMB is quite striking. I would suggest adding a comment concerning the fact that the GrIS is much closer to the historical range in 2030 than the AIS.*

That's a good point. The Antarctic SMB is very sensitive to temperature changes, since precipitation (snowfall) increases with increasing temperatures (through the Clausius-Clapeyron relationship). Since CESM2 is characterized by a relatively high climate sensitivity, the AIS atmosphere has already been warming in the period 1990-2014, which explains why the Antarctic SMB in the future simulations is clearly higher than the historical SMB from the beginning onwards. This is not as clear for Greenland, where snowfall and runoff can compensate one another, leading to a much smaller change in SMB.

To address the comment, we changed the text to:
"Figure 10b shows AIS SMB. In contrast to the GrIS, surface runoff plays only a minor role on the AIS (Figure A4) and the overall trend is dominated by increased snowfall with temperature, resulting in increased SMB. **A discernable departure from 1960-1999 values is obvious in all simulations due to end of 20th century warming**."

*9. Sect. 4.7 When discussing the ozone here, or in the conclusions section, it may be worth briefly mentioning that our view of the recovery of the ozone hole through column-integrated measurements may be somewhat biased when focusing exclusively on the high latitudes (e.g. Ball et al., 2018).*

*https://www.atmos-chem-phys.net/18/1379/2018/*

The review points to the paper by Ball et al., which focuses on latitudes outside the polar region. We agree that column integrated measures in mid and low latitudes can be misleading with regard to halogen induced ozone recovery. However, in high latitudes, column-integrated measures are still a very good measure for the recovery of the ozone hole. We slightly revise the text to point to the fact that we are focusing here on changes in high latitudes:

"These changes, in addition to the cooling of the surface and the troposphere, influence the strength of the sub-tropical and polar jets and therefore transport of stratospheric **airmasses, which results in changes of ozone**. In addition, stratospheric aerosols increase the aerosol surface area important for heterogeneous reactions. This leads to an enhanced activation of chlorine and therefore increased ozone depletion **in the polar stratosphere.** The effect of SAG was estimated to delay the recovery of the ozone hole by at least 40 years (Tilmes et al., 2008).

And add to the end of this section:

"Changes in stratospheric ozone in regions outside the SH polar latitudes as the result of SAG will be discussed in future studies."

*10. Sect. 5 Perhaps worth mentioning again the "peak-shaving" concept in the first paragraph as motivation for using the OS scenario?*

We agree and mentioned the peak-shaving scenario again in the text:

"The resulting overshoot in surface temperatures above the desired temperature targets **allows the application of a peak-shaving scenario** that requires limited SAG applications in time and amount, compared to steadily increasing injections needed for a high forcing scenario."

*11. Paragraph starting on l. 423: I find the paragraph to be poorly written and the overall message it carries unclear. Specifically:*
*l. 423 " to control for" ◊ "to achieve" or "to follow" or similar*
*l. 424 "can be done in" ◊ "can be done by"*

*ll. 424-425 Specify what these improvements are (i.e. reduced climate/bio-geophysical impacts, as opposed to improvements in actually meeting the global mean temperature target).*
*l. 427 "and are therefore provide a complete picture" ◊ "and therefore provide a more complete picture"*
*"Both limited applications of SAG and improved climate targets result in reduced climate impacts and risks, and are therefore provide a complete picture for studying impacts on society and ecosystems than much larger scale SAG applications." By "improved climate targets" do you mean following future scenarios of lower emissions? In any case, I do not think that lower emissions and lower SAG injections "provide a complete picture for studying impacts on society and ecosystems than much larger scale SAG applications". The completeness of the picture is not related in any way to the level of climate impacts and risks. Indeed, one may have a very complete picture of a high-risk scenario or a very incomplete picture of a low risk scenario.*

As pointed out more clearly above, we are aiming for impact relevant temperature targets of 1.5°C or 2.0°C. Furthermore, we are not only keeping global mean surface temperature at these levels, but also keep interhemispheric and pole-to-equator surface temperatures from changing. If SAG would be applied in this way, one would expect reduced side effects as compared to following unreasonably high injection scenarios or unreasonable setups. We propose that impact studies are more meaningful if they are performed based on scenarios that seem to be more reasonable than others. We adjusted the text in the manuscript to make this point clearer and improve the paragraph:

"In addition to reaching global surface temperature targets, the experiment requires **the achievement of** interhemispheric and pole- to-equator temperature targets, which can be done **by** using a feedback control algorithm to identify annual stratospheric injection amounts at 4 different latitudinal locations. For example, Kravitz et al. (2017) have shown several improvements in using the feedback controller to achieve the three temperature targets. Reaching global temperature surface targets **of 1.5°C or 2.0°C and keeping interhemispheric and pole-to-equator temperatures from changing has been shown to reduce global impacts,** including heatwaves, sea ice melting, **and large shifts in precipitation patterns**. **This scenario would therefore** reduce climate impacts and risks, and provide an improved basis for studying impacts on society and ecosystems **as compared to using unrealistic SAG applications based on a high greenhouse gas scenario. However, before impact studies are meaningful,** multi-model experiments are **required** to identify the range of outcomes and uncertainties. We therefore recommend including these experiments as a new testbed GeoMIP scenario for CMIP6."

*12. l. 435 "result in small differences in the amount of warming in high latitudes between a 1.5 °C and a 2 °C temperature target". Is this really the case? For the SH I agree, but for the NH high latitudes the difference between the OS1.5 and OS2.0 cases seems to be in the region of 0.5 °C. Perhaps providing exact numbers would help here.*

What we mean here was the "relative" warming of the high latitudes in each of these scenarios, not the absolute change. In Figure 4, changes are illustrated with regard to 2015-2025

conditions, which will result in a difference of 0.5C between the Geo 1.5C and Geo 2.0C case. We change the text to "**relative** amount warming".

*13. Formula A1: I am a bit puzzled by the notation. Having a summation over index j, without any j dependence in the summed terms would equal to multiplying these terms by k. Also, shouldn't the LHS of the equation have some dependence on "i"? I admit I am confused by this equation, and the individual terms/indices should be explained more clearly in the text.*

Thanks for pointing us to this mistake. We mistakenly used the wrong index and T[i] should have been T[j]

The correct formula should read:

$$S[k+1] = \hat{S}[k+1] + K_p(T[k] - T_{goal}) + K_i \sum_{j=0}^{k}(T[j] - T_{goal})$$

We also changed the text to explain the different quantiles:

"Where $S$ (the injection amounts in the next year) is the sum of the best estimate or feedforward value for year k+1, $\hat{S}[k+1]$ and a feedback correction. $K_p$ and $K_i$ are the proportional and integral control gains, whereby $K_p$ only reacts to the temperature error in the previous year ($T[k]$). $K_i$ is required to ensure zero steady-state error (the correction in response to the integrated error will continue to build as long as there is nonzero bias in the error). The summation in the integral term begins from the first year of injection ($j=0$) through to the year of simulation just completed."

*Other Remarks*

*l. 170 "Geo RCP-85 1.5" make sure this matches the label in Table 1 and its caption.*

Thanks, we consistently changed it to "Geo RCP85 1.5"

*l. 191 "could be differences" ◊ "could be due to differences"*

Thanks, this has been fixed.

*l. 198 "falling" ◊ "fall"*

Thanks, this has been fixed.

*l. 222 "increasing greenhouse gases" ◊ "increasing greenhouse gas concentrations"*

Thanks, this has been fixed consistently throughout the text.

*Table 1: Missing °C in the left-most column. Final entry reads "2."*

Thanks, we added °C for all the entities in the right panel (assuming that is what the reviewer means)

*l. 414 "and" ◊ "or"*

Thanks, this has been fixed.

*Sect. 4.4 Since the section only discusses data for SMB, it would be better to add "surface" in the title.*

We agree and changed it to "Ice sheet **surface** mass balance"

*l. 434 "three temperature targets" The temperature targets are only two, and one is considered for two different baseline scenarios.*

Here we are referring to the three temperature targets applying the feedback algorithm, namely, global surface temperatures, interhemispheric and pole-to-equator temperature gradients.

We are adjusting the text to clarify:

**"Applications of the feedback controller to achieve the three temperature targets in WACCM6, global surface temperatures, and interhemispheric and pole-to-equator temperature gradients,** result in small differences in the amount of warming in high latitudes between a 1.5°C and a 2°C temperature target. "

*l. 447 "illustrate" ◊ "illustrates"*

Thanks, this has been fixed.

*Caption Fig. 1 "methan" ◊ "methane"*

Thanks, this has been fixed.

*Fig. 1a Even though you specify the units in the title it is always good practice to label the axes too.*
We have added a label to the y-axis for panel a).

*Fig. 1 In all panels, the units (K) appear twice.*

We assume that the reviewer is pointing here to Fig. 5 and not Fig. 1, and have change the unites K to unites C and removed the unites in the title of each panel.

*Fig. 3 Caption "result" ◊ "results"*

Thanks, this has been fixed.

*Fig. 5 I struggle to identify the significant regions (to me, all regions in all panels appear to be "shaded in colour"). Also, what test was used to define the significance of deviations from climatology?*

There are white areas in the panel between the pink and light blue shades that are not colored. We used the *t*-test for significance and added in the caption:

"Regions shaded in color are significant with 95% confidence **based on the Student's *t*-test**."

*Fig. 7 "The shaded area is 1 standard deviation of 450 years pre-industrial control simulation" I am not sure I see the logic in this. Why add the PI SD to the SSP scenarios? If the aim is to provide a SD comparison, a table or some numbers in the text would be more indicated.*

In Figure 7, we show the two ensemble members (solid lines). On top, we plot the natural variability based on the pre-industrial control simulation, in order to indicate if differences are larger than the natural variably and if they therefore significant. In the revised manuscript, we have now applied the same method consistently for land NPP (Fig.7 and 8) and SSTs and ocean NPP (Figures 9, A3) and using for significance 1.96 times standard deviation from PI control (or 95% confidence of significance).

To clarify we added to the text:

"Fig. 7 shows the accumulated annual land NPP in different baselines and SAG scenarios. **The shaded areas around the curves illustrate the natural variability of NPP based on pre-industrial control conditions. Differences between the scenarios are considered to be significant if they lie outside the shaded area, as the case for SP5-85 and Geo SSP5-85 1.5, showing a slight reduction in NPP if SAG has been applied. The other scenarios do not show a significant difference.**"

*Fig. 8 Caption "pannel" ◊ "panel"*

Thanks, this has been fixed.

*Fig. 8 "Hatched regions are areas with changes within 1 standard deviation of 450 years pre-*

*industrial control simulation." It doesn't make much sense to show this unless it is discussed/commented upon in the paper.*

As for Figure 7, we added to the text:

"Figure 8 shows NPP anomalies between the three SAG scenarios and their baseline during 2060–2069. **Significance of results is assumed if the difference between the two simulations that are compared is larger than the natural variability, assuming the standard deviation (σ) in each model grid cell of the yearly means from the 499-year pre-industrial control run. An anomaly was considered significant when it was greater than 1.96 $\sigma$ (95\% confidence interval assuming normally distributed data)"**

*Fig. 9 See comment (7) above concerning the 95% level.*

Thanks for the commented we added: "assuming normally distributed data" to the figure caption.

*Fig. 11 Caption: "indicats" ◊ "indicates"*

Thanks, this has been fixed.

*Fig. 11 Caption: "from the one-member simulations." Don't all simulations have two members? Do the authors mean "the individual model realisations"?*

We thank the reviewer for catching this mistake, we reworded the sentence to: "A running mean over 5 years has been applied to the results."

*Figure A1 caption: it would be helpful to include these functional shapes in the description of the algorithm in the Appendix text.*

To address this comment, we have added the formulas of these functions into the text and refer to them in the Figure caption:

"The time-varying amount of cooling relative to the desired target was computed using the baseline simulations and fitted **with** a simple functional form. The desired feedforward was scaled from the values previously used values in the GLENS simulations. These had been estimated from earlier simulations, and so the feedforward estimates were somewhat different from what is needed in WACCM6. **The feedforward functions ($\hat{S}$) had to be fit to the different cases as illustrated in Figure A1 with $k$ being a function of years between the start of the injection and the end of the simulations:**

- SSP5-8.5, 1.5°C target: $\hat{S}[k] = 0.045(k - 2020)$

- SSP5-34-OS, 1.5°C target: $\hat{S}[k] = 0.045(k - 2020)$ for $k < 2050$, $\hat{S}[k] = 0.31 + 0.0487(k - 2020) + 0.000502(k - 2020)^2$

- SSP5-34-OS, 2°C target: $\hat{S}[k] = \max(\hat{S}_{1.5C}[k] - 0.5, 0)$

Figure A1 caption: "Top panel shows the fit to the desired temperature reduction for different cases **(see text for details)**.

*Fig. A2, A3 As above: how is significance computed?*

Please see comment to Fig.9.

*Fig. A4 Would it be possible to add a reference historical range as done in Fig. 10?*

Thanks for the suggestion. We added the historical range in Fig. A4.

**Reviewer 2**

***Suggestions for revision or reasons for rejection (will be published if the paper is accepted for final publication)***

*The authors have significantly improved the paper. The focus is now shifted to present and discuss the results of the simulations. A clearer motivation is given explaining the novelties and the relevance of the present study. A second set of experiments has been performed, which in my view is a major improvement as it gives some indication of the uncertainties coming from internal variability. Although the intention to establish a new testbed (and the related protocol) for model experiment has somewhat put more in the background in the present version, I believe that this study will motivate other groups to perform similar simulations. Not only in this respect the new appendix describing the feedback controller is very useful.*

*Overall, I believe the manuscript now provides sufficient new and valuable material to warrant publication. I have only found one (very minor) technical point:*

*Table 2 caption: meansures -> measures*
Thanks, we have fixed the typo.

[revised manuscript text omitted]

---

## Author Response (AR4)

Dear Editor,
We have already addressed all comments from the two referees. No additional changes have been made.
Best regards, Simone Tilmes